# Exploring varicella zoster virus proteome for construction and validation of a multi-epitope based subunit vaccine using multifaceted immunoinformatics approaches

Yassir A. Almofti[1☯*], Amna A. Ibrahim[2☯], Nuha A. Mahmoud[2], Abdelmajeed M. Elshafei[2], Nosiba Ibrahim[2], Ibrahim Albokhadaim[1], Saad Shousha[1], Ahmed O. Alameen[1], Nawal Elkhair[1], Mahmoud G. El Sebaei[1], Mahmoud Kandeel[1], Samir Alhojaily[1], Sheryar Afzal[1], Mohammed Ali Al-Hammadi[3], Ghada M[1], Ali Attiq[4], Yuan-Seng Wu[5]

1 Department of Biomedical Sciences, College of Veterinary Medicine, King Faisal University, Al-Ahsa, Saudi Arabia, 2 Department of Biochemistry, Faculty of Medicine and Surgery, National University/Sudan, Khartoum, Sudan, 3 Department of Microbiology, College of Veterinary Medicine, King Faisal University, Al-Ahsa, Saudi Arabia, 4 Discipline of Pharmacology, School of Pharmaceutical Sciences, Universiti Sains Malaysia, , Penang, Malaysia, 5 Department of Medical Education, School of Medical and Life Sciences, Sunway University, Subang Jaya, Selangor, Malaysia

☯ These authors contributed equally to this work.
* yamofti@kfu.edu.sa, yamofti99@gmail.com

## Abstract

Varicella zoster (VZ) is a viral disease caused by varicella zoster virus (VZV) that is related to alphaherpesvirus subfamily. VZV causes a neurotropic disease in humans. The aim of this work was to stimulate the human immune system by developing a multi-epitope vaccine based on five VZV surface proteins. Multiple immunoinformatics techniques were applied to assess B-cell and T-cell epitopes. The population coverage of each T-cell epitope was analyzed and the results showed high population coverage scores. The vaccine consists of 615 amino acids and it was antigenic and non-allergenic. The vaccine's hydrophilicity, stability, presence of aliphatic side chains, and thermal stability were all determined through an analysis of its physical and chemical characteristics. Also the vaccine demonstrated the least homology to human proteome (11%). Using the PSIPRED server and the Ramachandran plot, the vaccine's secondary and tertiary structures were predicted, enhanced, and validated. The structural errors were assessed using the ProSA web tool. Furthermore, the vaccine's solubility was higher than that of the *E. coli* proteins. Following immune simulation, there were significant amounts of T-cells, INF-γ, IL-2, and antibodies. Significant docking scores were obtained for each predicted epitopes and for the vaccine when docked to TLR4 chains. The TLR4-vaccine complex was highly stable according to molecular dynamic modeling. *In silico* cloning was performed to assess the vaccine's expression in the pET28a(+) vector, and the cloning results were efficient for translation. To ascertain the vaccine's effectiveness, *in vivo* and *in vitro* investigations, including clinical trials are required.

**Data availability statement:** All relevant data are within the paper and its Supporting information files.

**Funding:** This work was supported by the Deanship of Scientific Research, Vice Presidency for Graduate Studies and Scientific Research, King Faisal University, Saudi Arabia, under the Annual Research Track (KFU241682). The funder had no role in study design, data collection and analysis, decision to publish, or preparation of the manuscript.

**Competing interests:** The authors have declared that no competing interests exist.

## 1. Introduction

Varicella zoster virus (VZV) is an exclusively human neurotropic virus that belongs to the alphaherpesvirus subfamily [1–3]. Similar physiological and genomic characteristics were observed between VZV (the prototype neurotrophic human alphaherpesvirus) and herpes simplex virus type 1 (HSV-1). Both viruses are usually acquired in the early stages of development, when they are able to enter the ganglia and turn dormant [2,4]. Reactivation from dormancy causes the infectious virus to replicate and shed, ensuring the spread and transmission to an unsuspecting population [5,6].

Varicella is an acute, highly contagious viral disease with worldwide distribution. VZV has a very high transmission rate, mostly through airborne transmission. Morbidity and mortality rates are higher in immunocompetent children, adults, and the immunocompromised individuals compared to healthy children [7]. Strong seasonal patterns are evident in varicella, with peak incidence occurring in the winter and spring or during the cool, dry season [7,8]. There is considerable yearly variation in the incidence of varicella cases per 1,000 persons [7].

The genome of VZV is made up of linear double-stranded DNA and is roughly 125 kilobase pairs [9]. About 91% of the VZV genomes are made up of 74 open reading frames (ORFs) that make up the coding regions. It has been observed that the VZV genome is substantially conserved [10]. Varicella, often known as chickenpox, is caused by this virus and remains latent throughout the neuraxis in the cranial nerve ganglia, dorsal root ganglia, and autonomic ganglia [1–3,6]. VZV was shown to reactivate in old and immunocompromised people due to the reduction in VZV-specific cell-mediated immunity, causing shingles or herpes zoster [1,11–14]. This case is mostly characterized by post-herpetic neuraglia, meningoencephalitis, VZV vasculopathy, cerebellitis, myelopathy meningoradiculitis and eye damage. A reactivated VZV can also cause chronic radicular pain without rash. Moreover, VZV can cause all of the neurological disorders mentioned above without a rash [11–14]. More evidence connecting VZV to giant cell arteritis is rapidly emerging. For instance, approximately 95% of people worldwide have VZV infection and 50% will experience zoster by the time they are 85 years old or older and the neurological issues will continue to be a concern [15–18].

Antiviral drugs can be used to treat Varicella [4,19]. However, due to the low clinical benefit of these drugs in healthy individuals and their high cost, antiviral therapy is recommended only for those at high risk of severe varicella. The most effective method for treating VZV infection is the injection of anti-VZV antibodies. On the other hand, the most effective form of treatment is intravenous injection of acyclovir, which may need to be continued in cases of chronic sickness or in immunosuppressed individuals [1,19]. The most common side effect of herpes zoster medication is post-herpetic neuralgia (PHN), which is extremely resistant to therapy and causes excruciating, recurrent pain for at least three months after the rash [20,21]. Apart from post-herpes zoster (PHN), reactivation of VZV can result in a variety of neurological conditions, including meningoencephalitis, myelitis, zoster sine herpete (a rashless variant of herpes zoster), and Ramsay Hunt syndrome [22]. Currently, viral anterior uveitis (VAU) is treated with antiviral medications [21]. Nevertheless, there are no

consensuses or recommendations regarding which approach yields the greatest results and the least amount of ocular complications [23].

A live attenuated varicella vaccine, Vaccinax, is indicated for preventing varicella in infants and young children (age 12 months or older) [24]. This live attenuated vaccine contains a mixture of VZV genomes with varying polymorphisms and is the only human herpesvirus vaccine that has been developed. The innate antiviral barriers are reduced by mutations accumulated during tissue culture passage but do not affect VZV pathogenesis in T-cells or dorsal root ganglia (DRGs). During VZV pathogenesis, the viral–host interactions are modulated to avoid an overwhelming infection, which is beneficial for the virus since it ensures that this ubiquitous pathogen will continue to be transmitted and persist in the population [24]. Adverse effects of the VZV vaccine in healthy individuals are less and transient. However, vaccination resulted in a sore arm post-injection only in 20%–25%, and rashes similar to mild varicella infection in 5%. These rashes usually appeared a month post-immunization accompanied by mild fever [25].

*In silico* studies have expedited the time and obtained favorable results with lower expenses associated with laboratory pathogen studies. They have supplanted traditional culture-based vaccines [26,27]. Thus, the development of the novel vaccines by immunoinformatics methods may greatly reduce the incidence of varicella or chicken pox. The process of predicting epitopes often begins with analyzing the antigenic peptides' affinity for attaching to MHC molecules. Additionally, new potent vaccination protocols were developed as a result of the use of such resources and tools [28–32]. Despite these approaches aid in the creation of a vaccine specific to a population, the problem can also be reframed to produce a "universal vaccine," or a vaccine that provides the highest level of protection to every individual on the earth [33,34]. In this study, by analyzing the virus's surface proteins, a multi-epitope immunization against VZV infection was intended to be developed. The VZV comprises multiple proteins, but only the surface proteins were targeted and analyzed for epitopes prediction for vaccine formation.

## 2. Materials and methods

### 2.1. Flowchart of multi-epitope vaccine construction against VZV

A flowchart illustrating the sequential steps involved in the process of the multi-epitope vaccine construction against the VZV was depicted in Fig 1

### 2.2. Retrieval of the viral whole proteome

Uniprot at (https://www.uniprot.org/) is a high quality and freely accessible resource of proteins sequences [35]. Through an exploration of the uniprot database, the entire Human alphaherpesvirus 3 (varicella zoster viral genome-Dumas strain) with 73 proteins were retrieved in FASTA format and used for further analysis. The names, lengths, and entries of these proteins were provided in S1 Table.

### 2.3. Subcellular localization and transmembrane topology of virus proteins

Phobius server (https://phobius.sbc.su.se/index.html) is a combined transmembrane topology (TMH) and signal peptide predictor server [36]. It was exploited to examine the distribution of the viral protein at the subcellular localization level. Moreover, a multi-class SVM classification method called CELLO v.2.5 subCELlular Localization predictor at (http://cello.life.nctu.edu.tw/) was also utilized for subcellular localization [37]. Each virus protein was subjected to Phobius and CELLO v.2.5 servers for the localization probability score to be extracellular, outer membrane and periplasmic proteins. Proteins showed the best localization scores were used for epitopes prediction. Also topcons web server at (https://topcons.cbr.su.se/pred/reference/) that examines the combined membrane protein topology and signal peptide prediction was used for determination of the transmembrane helices (TMH) in the proteins [38].

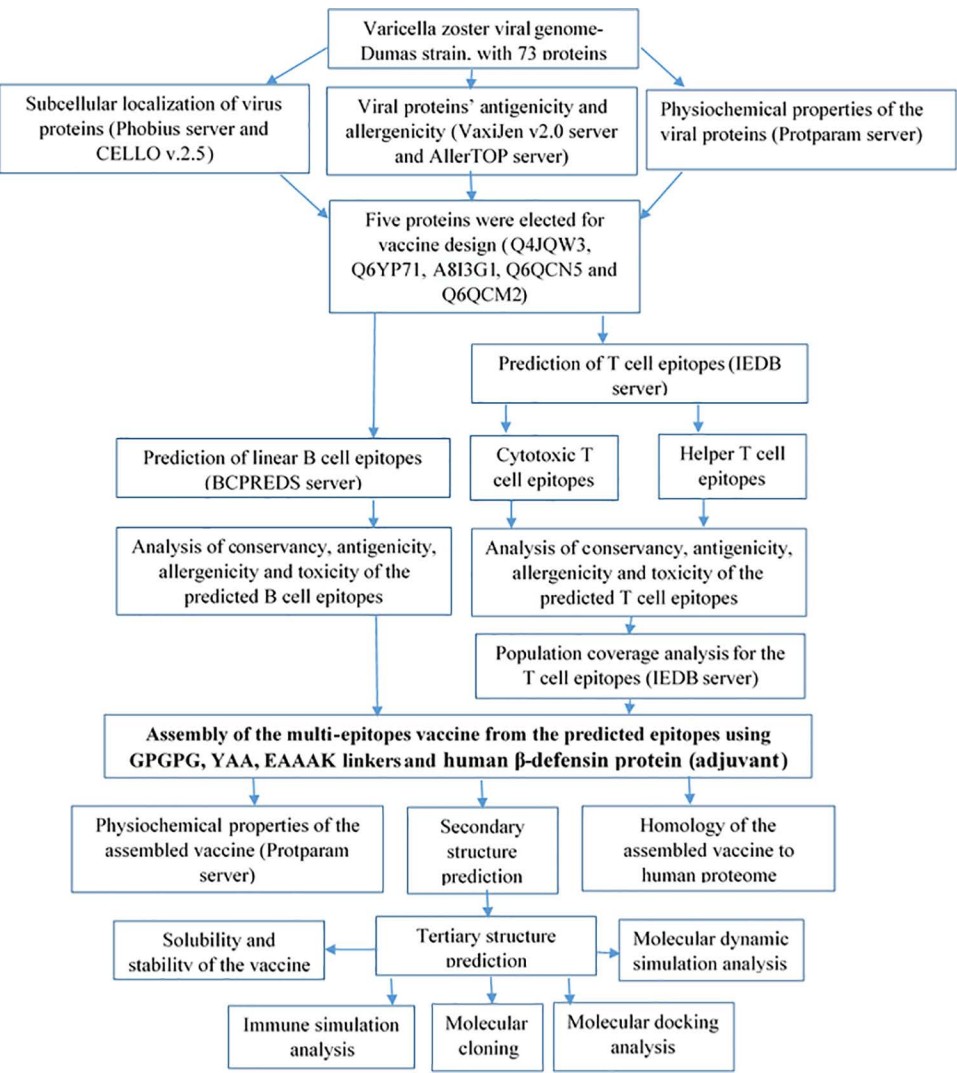

**Fig 1. The hierarchical steps used for computational vaccine design against VZV.** The workflow showed the essential phases such as the targeted proteins for epitope prediction, epitope identification, vaccine assembly, physiochemical analysis, the three dimensional structural modeling, molecular docking with immune receptors, molecular dynamics simulation, and *in silico* cloning.

## 2.4. The VZV proteins' antigenicity and allergenicity

The VaxiJen v2.0 server at (https://www.ddg-pharmfac.net/vaxijen/VaxiJen/VaxiJen.html) is considered as the first server for alignment-independent prediction of protective antigens [39]. This server classifies proteins based on the physicochemical properties of proteins, negating the requirement for sequence alignment. It was utilized to examine the strong antigenicity of VZV proteins using the default threshold of the server (0.4). Also AllerTOP server (https://www.ddg-pharmfac.net/AllerTOP/) is the first alignment free server for detecting the *in silico* prediction of allergens. The prediction relies on the physicochemical properties of examined proteins. Thus this server was used to assess the allergenicity of each of the virus proteins [40].

## 2.5. The biophysical and chemical characteristics of VZV proteins

The biophysical and chemical characteristics of the viral proteins were examined using the Expasy protparam service at (https://web.expasy.org/protparam/) [41]. Multiple metrics, such as length, molecular weight, extinction coefficient,

isoelectric point (pI), instability index, and grand average hydropathicity (GRAVY), were computed by the server for each protein. The analysis was performed to assess the suitability of the proteins for epitopes prediction.

## 2.6. The targeted proteins for vaccine's epitopes prediction

Based on the subcellular localization, biophysical and chemical characteristics, antigenicity and allergenicity of the viral proteins only five proteins were targeted in order to anticipate epitopes for vaccine formulation. These proteins were with the following uniprot entries: Q4JQW3, Q6YP71, A8I3G1, Q6QCN5 and Q6QCM2.

## 2.7. Prediction of the linear B-cell epitopes

The identification of B-cell antigenic epitopes directly aids in the development of vaccine components and immunodiagnostic reagents. Areas of the antigen surface that antibodies preferentially identify are known as antigenic epitopes. BCPREDS server for B-cell epitopes prediction at (http://ailab-projects2.ist.psu.edu/bcpred/) [42] was used to predict the antigenic B-cell epitopes. The server predicts Linear B-cell epitopes based on the physiochemical properties of the input proteins on a non-redundant databases such as Bcipep and Swiss-Prot databases. Also the server showed high epitope prediction accuracy when four combined amino acid characteristics (hydrophilicity, polarity, flexibility and exposed surface) were interfered in the prediction process. The length of the epitopes was set to 12-mers, and the epitopes prediction score was set to 0.51 in the server.

## 2.8. Prediction of the T-cell epitopes

There are two types of Major Histocompatibility Complex (MHC) molecules. The former is MHC Class I, which presents antigenic peptides (8–13 sequence length) to cytotoxic T-cell lymphocytes (CD8+TCR) in order to regulate non-self-intracellular antigens [43]. The latter is MHC Class II, which regulates extracellular antigens by presenting helper T-cell cells (CD4+TCR) with antigenic peptides (13–25 sequence length) [38]. Epitopes interacting against MHC-I and MHC-II molecules were predicted in this work.

### 2.8.1. Prediction of cytotoxic T-cell epitopes.
The immune epitope database (IEDB) at (http://tools.iedb.org/mhci/) was used to predict and analyze the epitopes that interact with MHC-I [44]. In this regard, each protein sequence was utilized as an input. The "NetMHCpan 4.1BA" method was selected as an epitope prediction method. Human was selected as MHC-1 source species. The epitope length was set to 9-mers and the human alleles such as HLA-A, HLA-B and HLA-C were examined for their interaction with each predicted epitope. Epitopes that bound to alleles at a score of ≤100 half-maximal inhibitory concentration (IC50) were selected for further investigation.

### 2.8.2. Prediction of helper T-cell epitopes.
Peptides attached to MHC-II molecules were analyzed using the IEDB MHC-II prediction tool at (http://tools.iedb.org/mhcii/) [45]. The prediction was performed with the human species and specific alleles HLA-DR, HLA-DQ, and HLA-DP using the NetMHCIIpanBA technique. The peptide length was set to 15-mers. While the core epitope within the peptide was set to 9mers. Each core epitope that coupled to an allele with an IC50 value of ≤1000 was selected for further investigation.

## 2.9. Strains retrieval, sequence alignments and epitopes conservancy analysis

All the retrieved strains from the five targeted proteins were obtained from the National Center of Biotechnology Information (NCBI) at (https://www.ncbi.nlm.nih.gov/protein/?term). Strain's retrieval for each protein was based on the sequence similarity and the length of the strain to the reference sequence of each protein. After strains retrieval, sequence alignment was performed using multiple sequence alignment (MSA) of Clustal W that integrated in the BioEdit tool, version (7.0.9.0) [46]. Based on the strain's alignment the conservancy of the predicted B- and T-cells epitopes was examined. Only epitopes that demonstrated 100% conservancy among the aligned strains from B-cell, MHC-I and MHC-II epitopes were further analyzed as vaccine's candidates.

## 2.10. Antigenicity, allergenicity and toxicity of the predicted epitopes

The predicted epitopes from the targeted proteins interacting against B and T lymphocytes were subjected for antigenicity prediction using the VaxiJen v2.0 server [39] with the default threshold (0.4). Only the antigenic epitopes were further analyzed for their allergenicity using AllerTOP server [40]. The ToxinPred3 server (https://github.com/raghavagps/toxinpred3) was utilized to evaluate the toxicity of epitopes that were determined to be both antigenic and non-allergenic epitope. Toxinpred server is an accurate *in silico* model for predicting the toxicity of therapeutic peptide. The server comprises huge number of toxic peptides from databases in addition to an equal number of SwissProt non-toxic peptides. The server predicts toxic peptides based on the sequence alignment, motif recognition, and machine learning analysis [47].

## 2.11. Population coverage

The population coverage of the antigenic, non-allergic and non-toxic epitopes was calculated for each epitope. Epitopes that interacted with MHC-I and MHC-II were subjected to population coverage analysis. The population coverage was analyzed using IEDB population coverage tool at (http://tools.iedb.org/tools/population/iedb_input) [48]. The population coverage analysis was initially performed for every single epitope that interacted with MHC-1 or MHC-II alleles against the whole world. Secondly, the population coverage was investigated based on area, country, and ethnicity by selecting one or two different groups from the whole word.

## 2.12. Assembly of the multi-epitope vaccine

Epitopes that expressed high population coverage scores were used to assemble the vaccine. Vaccine assembly was based on many linkers that assisted in enhancing expression, stability and folding of the vaccine molecule by separating the functional domains [49,50]. The epitopes of the B and T helper cells were fused together using a linker comprises glycine and proline amino acids (GPGPG Linker). Meanwhile, the linker containing tyrosine and alanine amino acids (YAA linker) was employed to fuse the T cytotoxic cell epitopes. To ameliorate the immunogenicity of the vaccine, the human β-defensin protein (uniprot Q5U7J2) was utilized as an adjuvant on the amino and carboxyl terminals [49]. The EAAAK linker was used to separate the adjuvant from the epitopes. Finally, in order to facilitate isolation and purification of the vaccine, a six histidine residues were tagged to the vaccine molecule [50].

## 2.13. Biophysical and chemical properties of the assembled vaccine

ProtParam analysis tool was exploited to analyze and compute many biophysical and chemical characteristics of the assembled vaccine including molecular weight, theoretical isoelectric point (pI), amino acid composition, atomic composition, extinction coefficient, estimated half-life, instability index, aliphatic index and grand average of hydropathicity (GRAVY) [41]. Also the number of the TMHs in the vaccine structure was assessed by topcons server [38].

## 2.14. Evaluation of the vaccine homology to human proteome

Using NCBI BLASTp (https://blast.ncbi.nlm.nih.gov/Blast.cgi?PAGE=Proteins), a protein blast for the vaccine protein was run to determine sequence similarity of the vaccine to the human proteome (taxoid: 9606). This blast was performed to avoid autoimmunity [51]. The vaccine's homology score to the human proteome must be less than 40% [52].

## 2.15. Prediction of the vaccine's secondary structure

The secondary structure of the vaccine candidate was predicted using the PSIPRED service (http://bioinf.cs.ucl.ac.uk/psipred/). The PSIPRED is based on the machine learning protocol and is designed to yield the three-state descriptions of the secondary structure (alpha helix, beta sheet, and coil) of the protein [53].

## 2.16. Prediction of the vaccine's tertiary structure, refinement and validation

The Iterative Threading ASSEmbly Refinement (I-TASSER) server at (https://zhanggroup.org/I-TASSER/) is a high-quality model prediction of the three dimensional structure (3D) and biological activity of proteins [54,55]. The I-TASSER was used to predict the 3D structure of the vaccine. The PDB files that provided by the I-TASSER server were structurally analyzed and the model with high RMSD was selected and further analyzed. The GalaxyRefine server (http://galaxy.seoklab.org/refine) [56] was utilized to improve (refine) the tertiary structure of the vaccine that was predicted by I-TASSER server. GalaxyRefine improves and refines the protein's local and global structure. This server is dependent on a refinement process via performing short molecular dynamics (MD) relaxations. The MD occurs after repeated side chain repacking perturbations. After the refinement of the vaccine structure, the vaccine model was initially validated and assessed by ProSA-web at (https://prosa.services.came.sbg.ac.at/prosa.php) [57]. The ProSA-web server determines the total quality score for a particular input protein PDB structure. Secondly, Ramachandran plot (https://saves.mbi.ucla.edu/) was utilized to validate the model via determining the stereo-chemical characteristics of the vaccine structure [58]. The Ramachandran plot provides a two-dimensional plot used for discriminating the conformational space of the phi and psi angles of the amino acids of a protein to determine the integrity and validity of the protein 3D structure model.

## 2.17. Solubility and stability of the vaccine construct

Protein–Sol server at (http://protein-sol.manchester.ac.uk) was utilized to determine the solubility and additional characteristics of the predicted vaccine [59]. The server exploited the solubility of E. coli proteins as a reference. The outcome is provided as a text file in graphical format and was utilized to predict the potential solubility of the vaccine [60]. Based on the mean solubility of E. coli proteins found in the experimental solubility dataset, proteins having a solubility score ≥0.45 are consequently expected to be soluble, and vice versa [59]. Regarding vaccine stability, disulfide links in the vaccine structure were engineered using the web-based application Disulfide by Design 2.0 (DbD2) at (http://cptweb.cpt.wayne.edu/DbD2/), which allows for disulfide engineering in proteins [61]. The high-mobility areas in the vaccine structure were determined by the server and were replaced by disulfide linkages. Therefore, by using the chi3 residue screening (between −87 and +97), B-factor value (ranged 6.950–17.410) and energy value less than 3.5, disulfide engineering was computed under the assumption that every amino acid residue pairs in the mobile regions were mutated to a cysteine-cysteine pairs.

## 2.18. Immune simulation

The C-ImmSim server at (http://150.146.2.1/C-IMMSIM/index.php) was used to simulate the immune response and immunogenicity of the vaccine [62]. The vaccine was administered at three time sets. The initial administration was at day one which represents the injection at time zero. Two injections followed the initial one at 90 and 180 days. The default settings were assigned to the remaining simulation parameters. The Simpson index (D) metric was calculated based on the simulation plot [62,63].

## 2.19. Molecular docking

Interactions between proteins during the docking process are essential to many biological processes. The intricate architectures of the docking interactions are important and had great impact on understanding the binding of the molecules [64]. In this regard, each predicted T-cell epitope was individually docked with MHC molecules to prove the high binding affinity of the epitopes to the MHC molecules. Initially, the PEP-FOLD 3.5 server, a de novo peptide structure prediction, was exploited to predict the structure of each epitope (ligand) as a PDBfile [65,66]. The MHC-1 (PDB: 3O3B) and MHC-II (PDB: 3TBP) were used as receptor molecules. The ligand-receptor interaction was performed using the advance method in the Cluspro protein-protein docking server [67,68]. Cluspro server is based on the removal of unstructured regions in

the protein, application of repulsion or attraction, calculating the pairwise distance restraints, building of homo-multimers, small-angle X-ray scattering (SAXS) consideration data, and finally detection of heparin-binding sites location. Moreover the server provides multiple energy functions based on the type of protein. Each energy function set provides ten centered models with highly populated clusters of low-energy docked structures. Finally, the docked molecules were visualized by PyMOL 3.1. Version 3.1.0 software. On the other hand, the vaccine molecule was docked with human Toll-Like Receptor 4 (TLR4) chains using the HADDOCK2.4 server (https://rascar.science.uu.nl/haddock2.4/) [69]. The HADDOCK2.4 refinement interface was exploited to provide the accurate cluster interaction. Thus, the vaccine PDB file ameliorated by Galaxyrefiner was considered as the ligand and submitted to the server with TLR4 (PDB: 4G8A) as a receptor. For TLR4, the docking process was performed with chain A and chain B separately. To view the graphical docking interaction between these molecules, the PDBsum server (https://www.ebi.ac.uk/thornton-srv/databases/pdbsum/) was used [70]. In addition, the PRODIGY web server at (https://wenmr.science.uu.nl/prodigy/) is a server focuses on the prediction of binding affinity in biological complexes and identification of biological interfaces from crystallographic one [71,72]. This server was used to provide the binding affinities of the docking clusters at 25°C.

### 2.20. Molecular dynamic simulation study (MD)

MD was performed to evaluate the stability and binding affinity of the vaccine construct with TLR4 in the docked complex. The simulation was performed using the Groningen Machine for Chemical Simulations program (Gromacs-2020.4) [73,74]. The solvated systems of the vaccine-TLR4 complex was exposed to energy minimization to the threshold of 100 kJmol−1nm−1. The simulation phase of unrestrained production 100 ns was applied at constant temperature conditions (300K) with the time step of 2 fs and trajectories stored at each 10 ps. The C-α atoms of the root mean square deviation (RMSD) for TLR4, the vaccine construct and vaccine-TLR4 complex were individually analyzed. Moreover the root mean square fluctuation (RMSF) in the side chain atoms, the radius of gyrations (Rg) and solvent accessible surface area (SASA) were also analyzed. The hydrogen bonds formed between the TLR4 and vaccine construct were analyzed and visually inspected in the intermittent trajectories [75].

### 2.21. *In silico* cloning

To guarantee the translation of the vaccine protein in the elected host, an *in silico* cloning was performed via Java Codon Adaptation Tool (JCAT) [76]. This server performs codon usage adaptation to the most sequences of the prokaryotic and some eukaryotic organisms. The JCAT server was used to convert the protein sequence of the vaccine into DNA sequence. In the JCAT, Codon Adaptation Index (CAI) best score is 1.0 but ≥0.8 is considered a good score as well [77]. The favorable GC content of a sequence ranged between 30%–70%. The BamH1 and Xho1 restriction enzymes sequences were added to the ends of the DNA sequence. The SnapGene restriction cloning module (version-2024) at (https://www.snapgene.com/guides/restriction-enzyme-cloning) was used to insert the DNA sequence into pET28a (+) vector between BamH1 and Xho1. The pET28a (+) vector was selected for the cloning process since it renowned for producing large amounts of protein. Also this vector has a strong T7 promoter that increases the vaccine protein production. Additionally the vector has multiple cloning site (polylinker) making it ideal for expression cloning.

## 3. Results

### 3.1. The targeted poteins for vaccine design

Based on the subcellular localization, antigenicity, allergenicity and the biophysical and chemical characteristics of the 73 proteins of VZV, only the tegument protein UL46 homolog (Q4JQW3), envelope glycoprotein B (Q6YP71), envelope glycoprotein C (A8I3G1), capsid protein (Q6QCN5) and capsid scaffolding protein (Q6QCM2) were used for epitopes prediction. The characteristics of these proteins were demonstrated in Table 1.

**Table 1. Physical and chemical properties, antigenicity and the subcellular localization of the five elected proteins.**

| Viral protein | Uniprot Entry | Molecular weight (Dalton) | Instability index # | Aliphatic Index | pI | No amino acids | Extinction Coefficient | GRAVY † | Vaxijen $ | Subcellular localization |
|---|---|---|---|---|---|---|---|---|---|---|
| Tegument protein UL46 homolog | Q4JQW3 | 74272.94 | 39.9 | 80.51 | 7.95 | 661 | 99865 | −0.395 | 0.4603 | OMP |
| Envelope glycoprotein B | Q6YP71 | 105347.36 | 40.73 | 78.31 | 8.81 | 931 | 115060 | −0.351 | 0.506 | OMP |
| Envelope glycoprotein C | A8I3G1 | 61353.05 | 45.41 | 81.34 | 8.73 | 560 | 75750 | −0.155 | 0.6612 | OMP/ P |
| Capsid protein | Q6QCN5 | 53971.4 | 35.95 | 83.6 | 9.00 | 483 | 70360 | −0.224 | 0.4882 | OMP/ E |
| Capsid scaffolding protein | Q6QCM2 | 66046.67 | 46.37 | 78.94 | 6.13 | 605 | 47790 | −0.336 | 0.5152 | OMP/ P |

#instability index <40 considered the protein stable.

pI: isoelectric point.

†GRAVY negative sign indicated the protein is hydrophilic.

$the threshold for the Vaxijen antigenicity was 0.4.

OMP: outer membrane protein. E: Extracelluar protein. P: Periplasmic potein.

## 3.2. VZV strains retrieval, sequence alignments and epitopes conservancy analysis

A different number of strains were retrieved based on the sequence similarity and the length of the retrieved strains to the reference sequence of the elected proteins. For instance, a total of 61 strains were retrieved for tegument protein UL46 homolog, 46 strains for envelope glycoprotein B, 172 strains for envelope glycoprotein C, 28 strains for capsid protein and 55 strains for capsid scaffolding protein. These strains were retrieved from the NCBI and further used for sequence alignment and epitopes conservancy. The tegument protein UL46 homolog and envelope glycoprotein B showed high level of conservancy among the retrieved strains in Bioedit software. While the other proteins showed moderate to low level of conservancy (S2 Fig). Only epitopes with 100% conservancy among all strains for each protein were used to construct the vaccine candidate.

## 3.3. B-cell epitopes prediction

Prediction of B-cell antigenic epitopes was obtained by using BCPREDS server. The tegument protein UL46 homolog, envelope glycoprotein B, envelope glycoprotein C, capsid protein and capsid scaffolding protein predicted 17, 26, 17, 13 and 12 linear conserved B- cell epitopes, respectively. The linear conserved epitopes were further tested for antigenicity, allergenicity and toxicity. Only epitopes that were proven to be antigenic, nonallergic and nontoxic were incorporated in the vaccine structure as B-cell epitopes and were listed in Table 2.

## 3.4. Cytotoxic T lymphocytes epitopes prediction

IEDB MHC-1 binding prediction tool predicted multiple epitopes interacting with MHC-1 molecule. The best epitopes which demonstrated antigenicity, nonallergic, nontoxic and highest population coverage scores were incorporated as candidates in the vaccine structure. These epitopes and their features were listed in Table 3.

## 3.5. Helper T lymphocytes epitopes prediction

IEDB MHC-II binding prediction tool predicted thousands of epitopes from each protein. The best epitopes which demonstrated high antigenicity, non-allergic, nontoxic and demonstrated high population coverage scores were incorporated in the vaccine structure. These epitopes and their features were listed in Table 4.

## 3.6. Population coverage analysis

The population coverage was calculated for the cytotoxic and helper T-cells epitopes that interacted with MHC-1 and MHC-II alleles, respectively. As shown in Tables 3 and 4, the epitopes showed high allelic interaction to the MHC alleles

**Table 2. The predicted B-cell epitopes, their antigenicity, allergenicity and toxicity from the five targeted proteins.**

| Protein | Epitope | Score | Start | #Vaxijen Antigenicity (0.4) | Allergenicity | Toxicity |
|---|---|---|---|---|---|---|
| **Tegument protein UL46 homolog** | GYACWGDGGLND | 0.978 | 365 | 1.2678 | Probable non-allergen | Nontoxin |
| | ISVLWRKEEWRD | 0.978 | 321 | 1.1311 | Probable non-allergen | Nontoxin |
| | GSPIGTGIGNLE | 0.977 | 471 | 1.2019 | Probable non-allergen | Nontoxin |
| **Envelope glycoprotein B** | ERQESKARKKNK | 0.999 | 889 | 1.0377 | Probable non-allergen | Nontoxin |
| | LDYSEIQRRNQM | 0.963 | 712 | 1.2840 | Probable non-allergen | Nontoxin |
| | ESLQVEPTQSED | 0.832 | 79 | 1.0063 | Probable non-allergen | Nontoxin |
| **Envelope glycoprotein C** | DGYPKKFPPFSA | 0.867 | 505 | 0.6644 | Probable non-allergen | Nontoxin |
| | TVTTYYRPNITV | 0.848 | 317 | 0.8429 | Probable non-allergen | Nontoxin |
| **Capsid protein** | DLTIGPRFGGLN | 1.000 | 52 | 2.1055 | Probable non-allergen | Nontoxin |
| | NPTSIGNPQVTI | 0.986 | 97 | 1.1465 | Probable non-allergen | Nontoxin |
| **Capsid scaffolding protein** | DRWDVVAKRRRE | 0.998 | 214 | 1.6350 | Probable non-allergen | Nontoxin |
| | SSQNTTSTPHTD | 0.997 | 556 | 1.6350 | Probable non-allergen | Nontoxin |

#Vaxijen antigenicity was the default threshold of the server (epitope antigenicity greater than 0.4 is antigenic).

**Table 3. The predicted T cytotoxic cells epitopes and their population coverage from the five proteins.**

| Protein | Epitope $ | Start | End | Length | #Vaxijen Antigenicity (0.4) | Population coverage |
|---|---|---|---|---|---|---|
| **Tegument protein UL46 homolog** | LAATTASAL | 307 | 315 | 9 | 0.7359 | 99.99% |
| | NKKSKSTVL | 168 | 176 | 9 | 0.7323 | 88.31% |
| **Envelope glycoprotein B** | TSSVEFAML | 509 | 517 | 9 | 1.1867 | 100.00% |
| | FGALAVGLL | 787 | 795 | 9 | 1.2303 | 100.00% |
| | TSRLTGLAL | 906 | 914 | 9 | 0.9287 | 99.96% |
| **Envelope glycoprotein C** | KFPPFSAVY | 510 | 518 | 9 | 1.3215 | 100.00% |
| | IQINLILTI | 4 | 12 | 9 | 1.0248 | 99.98% |
| | VISEHSITV | 310 | 318 | 9 | 1.4874 | 99.98% |
| **Capsid protein** | RFIQIGNGL | 29 | 37 | 9 | 1.2060 | 99.90% |
| | IQIGNGLHM | 31 | 39 | 9 | 1.1915 | 99.97% |
| | AYVTSLSFI | 193 | 201 | 9 | 1.0233 | 99.92% |
| **Capsid scaffolding protein** | RGPFFLGIV | 69 | 77 | 9 | 1.3159 | 99.93% |
| | TRDPRISIL | 536 | 544 | 9 | 1.4309 | 99.95% |

$: the predicted epitopes were probable non-allergen and non-toxin.

#: Vaxijen antigenicity was the default threshold of the server (epitope antigenicity greater than 0.4 is antigenic).

when analyzed against the whole world. In addition, Fig 2 demonstrated the population coverage based on area, country, and ethnicity by selecting one or two different countries from each area to cover the whole word. These results indicated that the epitopes of the vaccine could cover high population around the world.

## 3.7. Biophysical and chemical characteristics of the proposed vaccine

Twelve linear B-cell epitopes, thirteen T cytotoxic cell epitopes and eleven T helper cell epitopes were incorporated in the vaccine structure. In addition, the vaccine included adjuvant, linkers, and His-tag. Thus a total of 615 amino acids incorporated in the vaccine structure (Fig 3). The vaccine construct was antigenic (0.6228) and nonallergenic in Vaxigen and Allertop servers, respectively. Furthermore, the vaccine protein showed no signs of TMH. Table 5 listed the biophysical and chemical characteristics of vaccine construct.

**Table 4. The predicted T helper cells epitopes and their population coverage from the five proteins.**

| Protein | Epitope $ | Peptide | Start | End | Length | #Vaxijen Antigenicity(0.4) | Population coverage |
|---|---|---|---|---|---|---|---|
| Tegument protein UL46 homolog | AASKRYTPL | YYAQVLAASKRYTPL | 434 | 448 | 15 | 1.0643 | 99.34% |
| | AQVLAASKR | KPTKYYAQVLAASKR | 430 | 444 | 15 | 0.9505 | 99.98% |
| Envelope glycoprotein B | FSSIHPNAA | RFRRFFSSIHPNAAA | 23 | 37 | 15 | 0.8486 | 100.00% |
| | LSTGDIIYM | YDSFGLSTGDIIYMS | 290 | 304 | 15 | 0.8030 | 100.00% |
| Envelope glycoprotein C | FVNMQSSCP | QNKRFVNMQSSCPTS | 478 | 492 | 15 | 1.0183 | 97.14% |
| | INLILTIAC | KRIQINLILTIACIQ | 2 | 16 | 15 | 1.0210 | 96.72% |
| Capsid protein | KRTETASIQ | VPPKRTETASIQVTP | 69 | 83 | 15 | 1.0550 | 100.00% |
| | IQVTPRSIV | ASIQVTPRSIVINRM | 77 | 91 | 15 | 1.5821 | 100.00% |
| | FIQIGNGLH | NNRFIQIGNGLHMTY | 27 | 41 | 15 | 1.0035 | 100.00% |
| Capsid scaffolding protein | GVVAKLQQE | QHEELAGVVAKLQQE | 434 | 448 | 15 | 0.8366 | 99.34% |
| | AERGIDLQT | FMNALAAERGIDLQT | 430 | 444 | 15 | 0.9619 | 99.98% |

$: the predicted epitopes were probable non-allergen and non-toxin.

#Vaxijen antigenicity was the default threshold of the server (epitope antigenicity greater than 0.4 is antigenic).

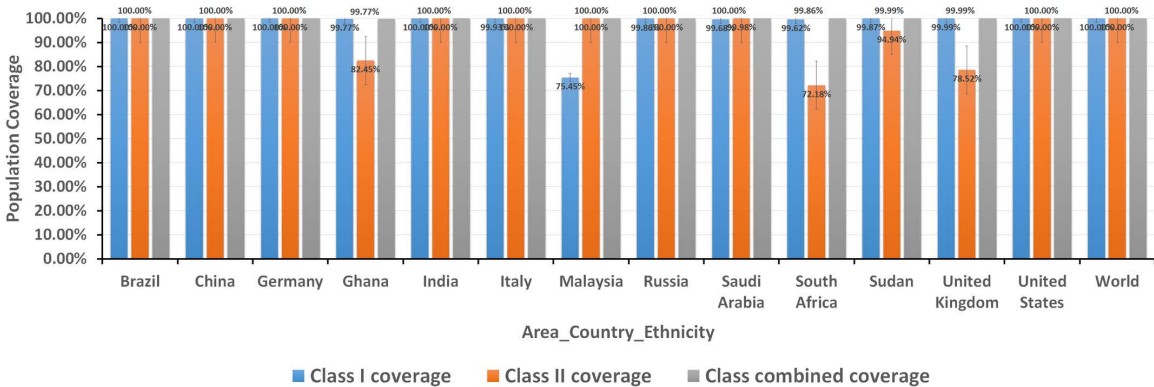

**Fig 2. The population coverage analysis based on the area, country and ethnicity.** The epitopes showed high population coverage when interacted with MHC-1 and MHC-II alleles of area, country and ethnicity groups. The combined MHC-1 and MHC-II alleles showed 100% coverage in all areas and countries and the result score was shown on the upper of each graph.

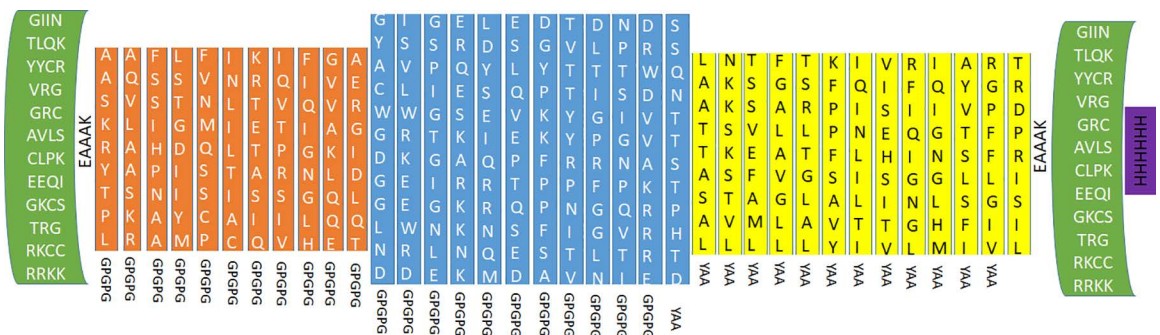

**Fig 3. Showed the vaccine sequence. The human β-defensin-3 (green color) was used as an adjuvant at N- and C-terminals and separated by the EAAAK linkers.** The T helper (red color) and B-cell epitopes (blue color) were linked by GPGPG linkers, while T cytotoxic epitopes (yellow color) were linked by YAA linkers. A 6-His tag (purple color) was added at the C- terminal.

**Table 5. The biophysical and chemical characteristics, antigenicity and the number of the predicted trans-membrane helices of the vaccine protein.**

| Characteristics | Measurement |
|---|---|
| Molecular weight (Dalton) | 63602.51 |
| #Instability index | 39.78 |
| Aliphatic index | 70.85 |
| pI | 9.69 |
| Number of amino acids | 615 |
| Extinction coefficient | 61615 |
| $Vaxijen antigenicity | 0.6228 |
| †GRAVY | −0.309 |
| TMHs | 0 |

#instability index <40 considered the protein stable. pI, is the isoelectric point. **$:** the threshold for the Vaxijen antigenicity is 0.4. **†**GRAVY: negative sign indicated the protein is hydrophilic. THMs: Transmembrane helices.

## 3.8. BLAST homology assessment

Homology between the sequence of the vaccine and the host proteome sequence demonstrated that the query coverage of the vaccine protein showed only 11% homology to human proteins. This result showed that the predicted vaccine would not implicate in autoimmunity diseases to the host.

## 3.9. Secondary structure of the vaccine construct

As shown in Fig 4 the secondary structure prediction of the vaccine construct showed 23.74% alpha helix, 4.72% beta turn, 19.84% extended strands and 51.71% random coiled.

## 3.10. Tertiary structure prediction, refinement and adaptation of the vaccine construct

The three-dimensional structure (3D) of the vaccine was ascertained using I-TASSER sever (Fig 5a). The refinement of the vaccine structure showed enhancement of the vaccine structure quality (Fig 5b). The ProSA results showed that a Z-score of −3.86 for the vaccine's overall model quality. This result indicated a reasonable model of the vaccine structure (Fig 5c). The Ramachandran plot showed that the number of residues in most favoured and additional regions was 92.8%, in generously allowed regions was 2.5% and in disallowed regions was 4.7% (Fig 5d). This result greatly enhances the stability of the vaccine 3D structure.

## 3.11. Solubility and stability of the vaccine construct

Fig 6a demonstrated the solubility of the vaccine construct in terms of QuerySol (scaled solubility value). The solubility of the vaccine was 0.580 based on *E. coli* proteins solubility as a reference (Fig 6b). For the stability of the vaccine construct, residues in the highly mobile region of the protein sequence (wild type) were mutated with cysteine to perform disulfide engineering (Fig 6c). A total of 61 pairs of amino acid residues were shown to be probably forming disulfide bonds. Among them only six regions were evaluated to form disulfide bonds and were replaced by cysteine residues. These six residue pairs were $ALA_{540}$-$GLU_{592}$; $ASN_{281}$-$GLY_{340}$; $ALA_{563}$-$GLN_{122}$; $GLY_{162}$-$ARG_{212}$; $VAL_{193}$-$GLY_{236}$ and $ARG_{230}$-$SER_{248}$ (Fig 6d).

## 3.12. Immune simulation

In the immune simulation, the obtained immunological simulation results were matching the real immune responses. This was demonstrated by a discernible rise in the antigens concentration and a corresponding decrease in the primary,

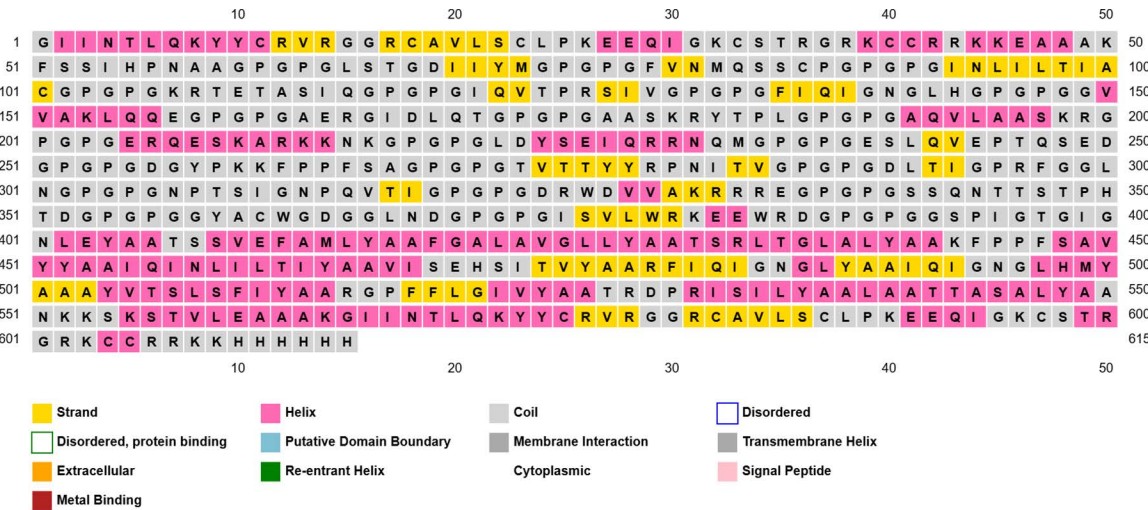

**Fig 4. Graphical representation of secondary structure features of the final vaccine sequence.** The sequence of the amino acids in the vaccine sequence and the location of alpha helixes and beta turns and coils were shown in the figure.

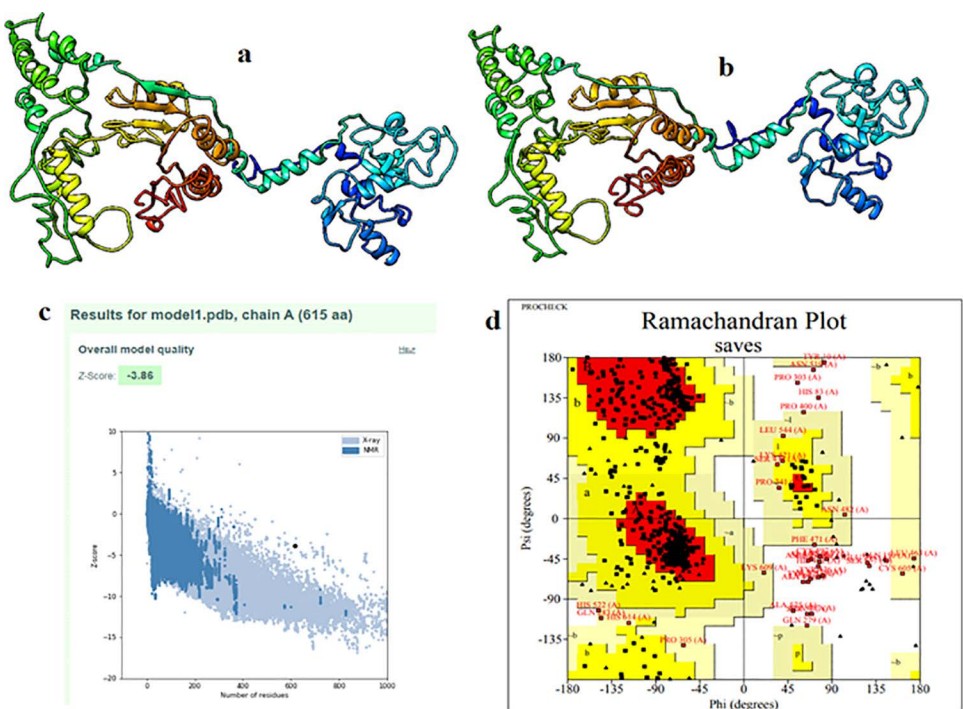

**Fig 5. (a): The 3D model of the vaccine construct obtained after homology modelling on I-TASSER.** (b): The 3D model was refined in galaxyrefiner and (c): ProSA-server, giving a Z-score of −3.86. (d): The validated refined model was assessed by Ramachandran plot analysis that demonstrated the majority of the residues was in favoured and allowed regions. While least residues were in the disallowed (outlier) region.

secondary, and tertiary immune responses (Fig 7a). Also the level of cytokines and interleukins (IL), mainly IL-2, was compatible with the Simpson index (D) (Fig 7b). Along with leukocyte growth factor, the growing diversity measure over time is regarded as a risk indicator. As a result, diversity showed decreasing value. Additionally, the rise in IgM was a

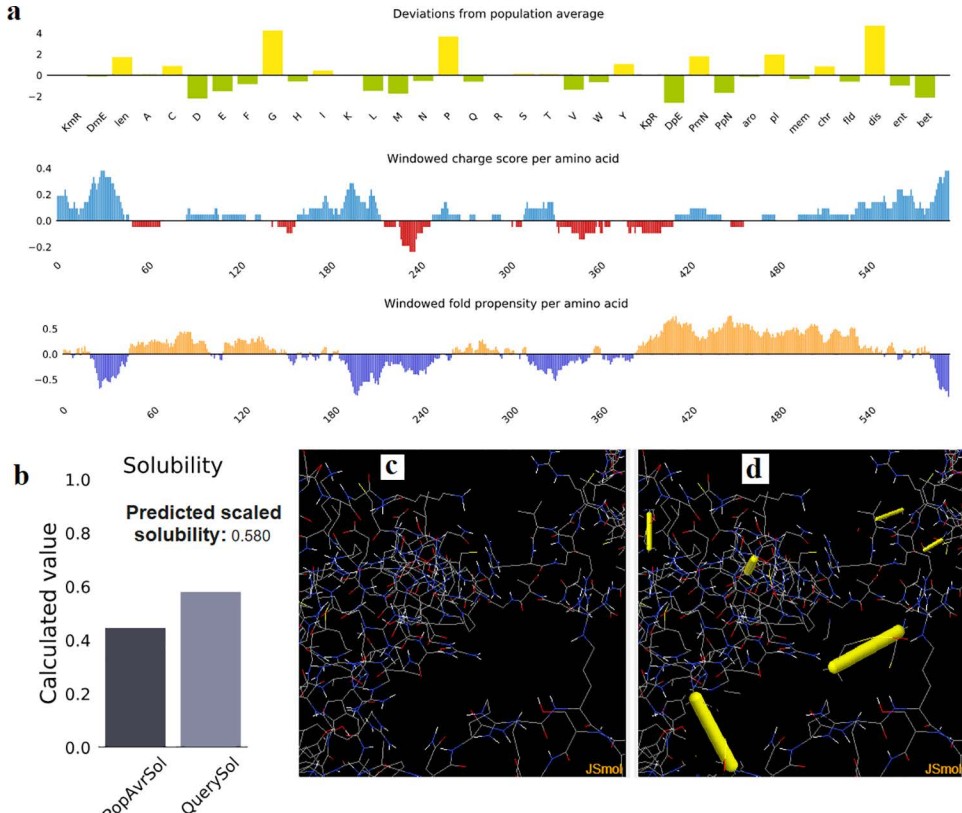

**Fig 6. (a): The deviation of the population average of the vaccine from the population average for the experimental dataset (PopAvrSol) as well as the windowed charge score and the windowed fold propensity per amino acid of the vaccine.** (b): Solubility of the vaccine construct was 0.580 compared to 0.45 of the *E. coli*. (c) The stability of the vaccine construct before engineering of the disulfide bonds (original form) and (d): the mutant form, with six disulfide bonds (golden sticks).

characteristic of the main reaction, whereas an increase in the B-cell population and the quantity of antibodies was a noticeable outcome of the secondary and tertiary responses (Fig 7c). This demonstrated how immunological memory was formed and how the antigen was quickly eliminated after new exposures. Additionally, high response levels were seen in the T-cytotoxic (TC) (Fig 7d) and T-helper (TH) (Fig 7e) lymphocyte populations, which corresponded with the establishment of memory. Throughout the exposure, the natural killer cell count remained high (Fig 7f).

### 3.13. Molecular docking

The epitopes that entered in the synthesis of the vaccine were initially docked against the MHC molecules. Fig 8 showed the docking of the cytotoxic T-cell epitopes with the MHC-I molecule. The binding of the epitopes showed strong binding energies (Table 6) that further strengthened by hydrogen bonds between the docked molecules. Also Fig 9 showed the docking of the helper T-cell epitopes with the MHC-II molecule. The binding of the epitopes demonstrated strong binding energies (Table 6) that further confirmed by hydrogen bonding between the docked molecules.

The vaccine construct was docked against TLR4 (chain A and chain B) using the HADDOCK2.4 server. Docking of the vaccine construct with TLR4 chain A provided eight structures with two clusters representing 4% of the water-refined models. The RMSD from the overall lowest-energy structure and the Z-score of the docking were 34.7 +/-1.0 and −1.0, respectively. This result demonstrated the strong binding between the molecules as indicated by the presence of 21 hydrogen bonds and 3 salt bridges (Fig 10 and Table 7).

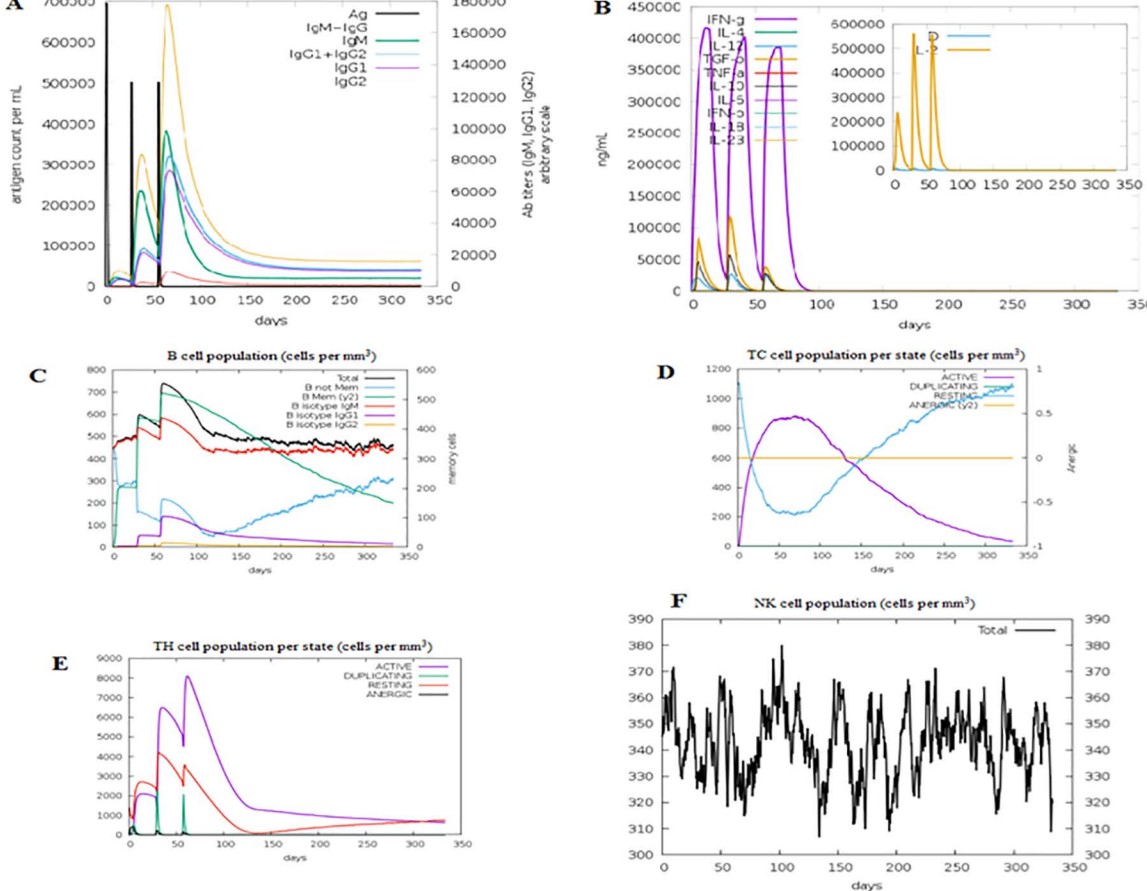

**Fig 7. The immune simulation of the predicted vaccine after the three injections.** (a): Antibodies production in response to antigen injections (antibodies were shown as different colored peaks and the antigen was shown in black color). (b): The induced cytokines secretion and the IL-2 level with the measure of diversity. (c): Showed the memory, not memory and the isotypes of B-cell populations. (d): Showed the active T-cytotoxic (TC) cell populations. (e): Showed the active T-helper (TH) cell populations. In (d, e): The resting state demonstrated the cells not provided with the antigen (vaccine). The anergic state demonstrated tolerance of the T-cells to the antigen due to repeated exposures. (f): Natural killer cell populations.

On the other hand, docking of the vaccine with TLR4 chain B provided fourteen structures with two clusters representing 7% of the water-refined models. The calculation of the RMSD from the overall lowest-energy structure and the Z-score of the docking showed 30.7 +/- 0.1 and −1.0, respectively, indicating the strong molecules binding. This strong binding was obvious by presenting 14 hydrogen bonds and 2 salt bridges between the molecules (Fig 11 and Table 8).

The calculation of the binding affinity between the docked molecules was provided by PRODIGY server. The binding affinity is expressed as free energy change (ΔG) and dissociation constant (Kd) illustrating the real interaction of the docked molecules within the cell. The binding affinity results of TLR4 chains and the vaccine complexes were provided in Table 9. The results indicated that the docked molecules were energetically viable.

### 3.14. MD simulation

Upon the molecular dynamic simulation, the RMSD was measured individually for the TLR4, vaccine and the vaccine-TLR4 complex. As shown in Fig 12 the RMSD simulation of each molecule reached the equilibrium rate of the system with sufficient time of simulation. Moreover, least RMSD fluctuations plot showed the stability of the vaccine-TLR4 complex in dynamic state. Therefore fluctuation in each molecule was analysed via RMSF. The RMSF fluctuation of vaccine molecule

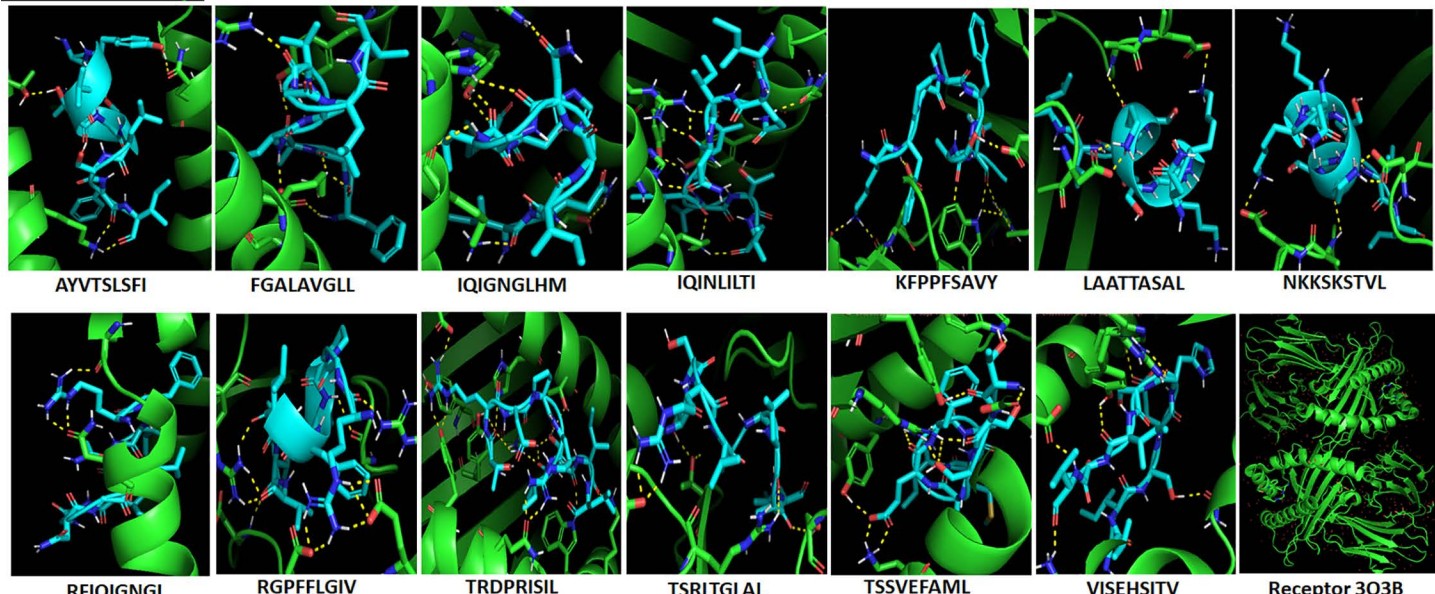

**Fig 8. Docking of the cytotoxic T-cell predicted epitopes with MHC-I molecule.** The epitopes were shown in cyan color and the receptor in a green color. The hydrogen bonding between each epitope and the receptor were shown in yellow dots. Each epitope sequence was provided in the figure as well as the complete structure of the receptor.

**Table 6. The table provided the lowest binging energies of each epitope when docked against the MHC-I and MHC-II.**

| MHC-I | | MHC-II | |
|---|---|---|---|
| **Epitopes** | **Binding energy score** | **Epitopes** | **Binding energy score** |
| VISEHSITV | −612.2 | LSTGDIIYM | −720.2 |
| TSSVEFAML | −729.2 | KRTETASIQ | −531 |
| TSRLTGLAL | −638.6 | IQVTPRSIV | −758.6 |
| TRDPRISIL | −672.5 | INLILTIAC | −709 |
| RGPFFLGIV | −759 | GVVAKLQQE | −581.3 |
| RFIQIGNGL | −774.7 | FVNMQSSCP | −720.3 |
| NKKSKSTVL | −486.8 | FSSIHPNAA | −724.8 |
| LAATTASAL | −784.9 | FIQIGNGLH | −718.7 |
| KFPPFSAVY | −803.7 | AQVLAASKR | −596 |
| IQINLILTI | −859.7 | AERGIDLQT | −685.7 |
| IQIGNGLHM | −773.8 | AASKRYTPL | −690.2 |
| FGALAVGLL | −753.9 | | |
| AYVTSLSFI | −694 | | |

was slightly higher than that showed by TLR4. This might be attributed to the increased ratios of coils and turns in vaccine structure. The radius of gyration (RG) of the TLR4 and the vaccine showed the compactness and closeness of these molecules forming stable structure. Also the SASA was remarkably good in the TLR4 and vaccine structures during the simulation process (Fig 13) and the stability of the molecules was further strengthened by the hydrogen bonds between the vaccine and TLR4. Thus, taken together, the vaccine construct and TLR4 complex showed good stability in solvated dynamic state.

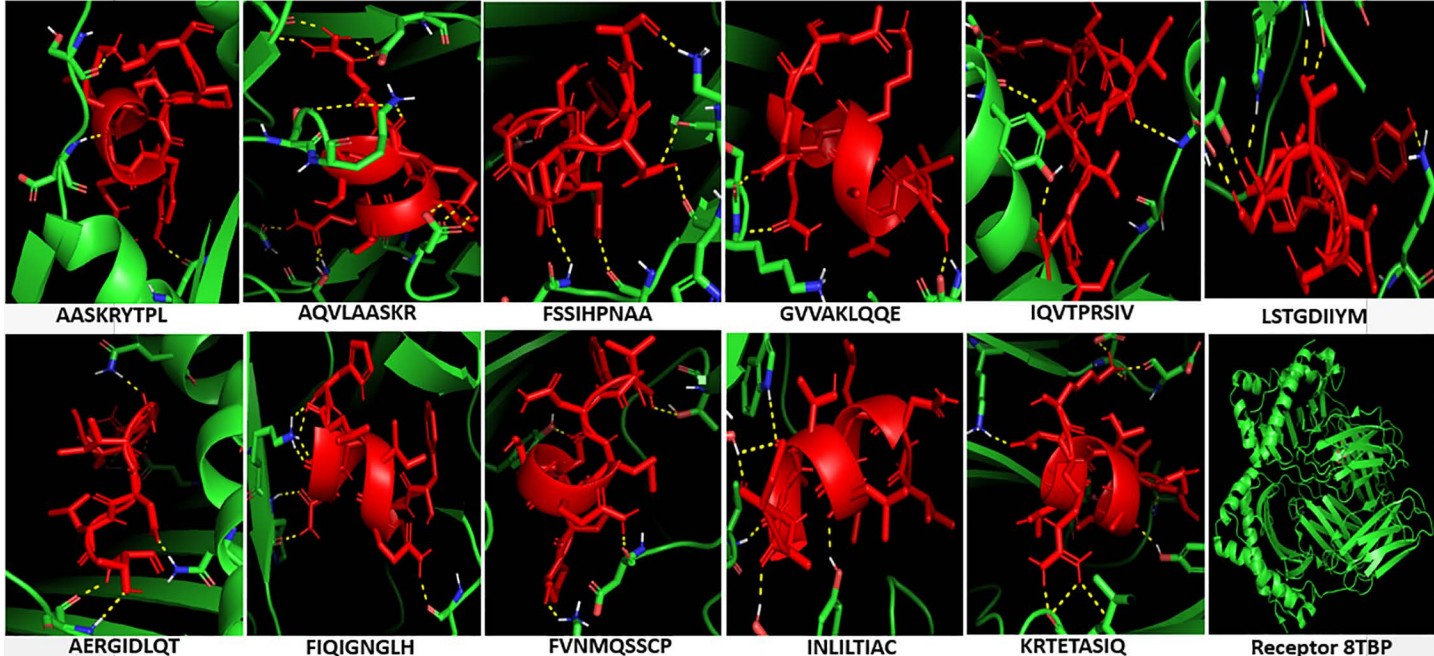

**Fig 9. Docking of the helper T-cell predicted epitopes with MHC-II molecule.** The epitopes were shown in red color and the receptor in a green color. The hydrogen bonding between each epitope and the receptor were shown in yellow dots. Each epitope sequence was provided in the figure as well as the complete structure of the receptor.

### 3.15. Codon adaptation and *in silico* cloning

The protein sequence of the vaccine was reversed translated into DNA sequence. Codon adaptation index values (CAI-Value) of the improved DNA sequence was 0.9199. This result indicated the higher proportion of most abundant codons. The GC-content of the improved sequence was 51.58%, indicating favourable GC content. Fig 14, showed that DNA sequence was cloned into pET28a (+) vector typically at the multiple cloning site (MCS) of the vector after fusing the BamHI and Xho1restriction enzymes cutting sites sequences to the vicinities of the DNA sequence.

## 4. Discussion

The aim of this study was to predict a multi-epitope vaccine candidate against VZV. The conventional vaccines development demonstrated multiple drawbacks as multiple organisms difficult to grow and culture and some demonstrates problems in the process of vaccine attenuation [78]. Moreover, some organisms may revert to more virulent strains resulting in threats and adverse events to human life [78]. Therefore, the shift to a new era of effective, safe, more adaptable, cost-efficient vaccines became of great importance [79,80]. *In silico* vaccine development is considered as a transformative step in the world of vaccine design due to the speed, adaptability and specificity of vaccine production. However, the *in silico* vaccine design faces several limitations, such as the discrepancies between predicted and actual immune responses across genetically diverse human populations. The *in silico* vaccines may fail to elicit strong immune responses in genetically diverse groups, particularly in populations from underserved regions. Also the prediction tools often focus on binding affinities of epitopes to HLA molecules but may overlook antigen processing and peptide translocation mechanisms that varies among individuals. This results in discrepancies between predicted and actual immune responses. Moreover the predicted epitopes might cause cross-reactive immune responses leading to immunopathology instead of protection. However integrating *in silico* predictions with functional

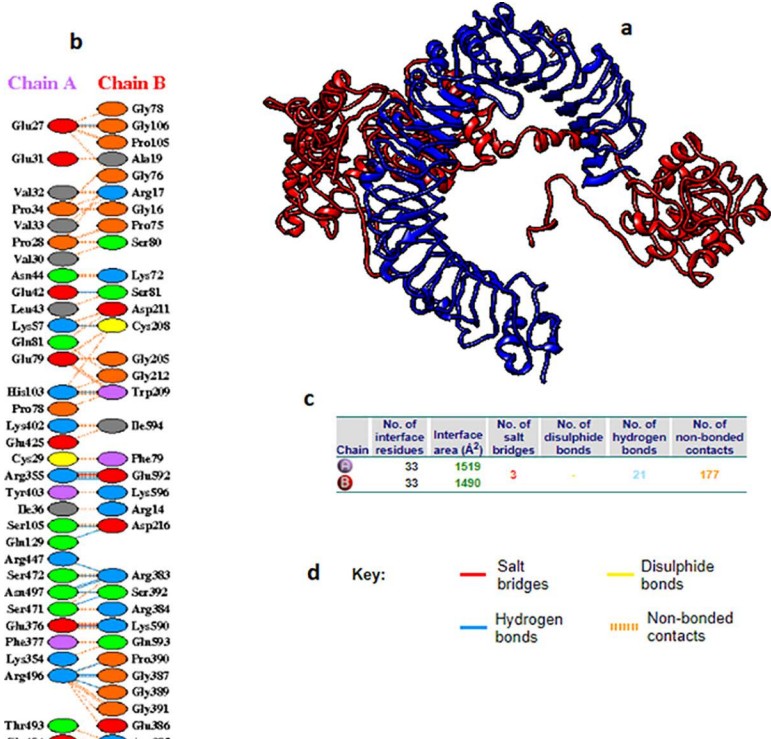

**Fig 10. (a) Docking of the vaccine construct (red colored) with the TLR4 chain A (blue colored).** (b) Residue interactions across the interface colored by residue type. The vaccine residues were presented as chain B and the TLR4 as chain A. (c) The interface statistics showing the number of bonded and unbonded residues. (d) The key showing the interacted bonds based on the color.

assays, empirical validation and inclusive datasets would enhance vaccine efficacy and safety across diverse populations [81].

Currently, the VZV lacks an effective vaccine that stop the VZV infection. Thus the need for an effect vaccine is crucial. In this study five proteins were used to construct the vaccine. The cellular localization of these proteins is very important to be recognized by the immune system [82]. Among the 73 proteins of VZV, only five proteins demonstrated high score as extracellular, outermembrane and/or periplasmic proteins. The surface proteins were selected since they were the first proteins to face the immune cells of the host [83]. The five proteins were selected for vaccine construction and their biophysical and chemical characteristics were investigated to assess their suitability for epitopes prediction.

In this study, the most suitable conserved epitopes were predicted as vaccine candidate and were shown to be antigenic, elicit an immune response, not causing allergenicity and were nontoxic epitopes. The T lymphocytes proposed epitopes were investigated for the global population coverage against the whole world, assessing their potential to induce immune response against these epitopes. Also the epitopes showed excellent population coverage percentages when analyzed by the area, country and ethnicity. These results indicated the vaccine molecule might cover large population and could act as a universal vaccine. However, it was reported that exclusion of some rare alleles and the genetic diversity in specific ethnic groups might affect the population coverage prediction process. Additionally, the exclusion of epitopes that are weak but biologically relevant binders could result in incomplete population coverage predictions [81].

During the process of vaccine design, short peptides were used as linkers between the B- and T-cells epitopes. Theses linkers were shown to provoke minimal junctional immunogenicity between the epitopes as previously described [84–90] and to reach a high level of expression and improve bioactivity of the vaccine [90,91]. However, the structural flexibility and

**Table 7. Atom−Atom Interactions list of the TLR4 chain A and the vaccine interface showing the hydrogen and salts bonding residues. Atom 1 represented TLR4 chain A and atom 2 represented the vaccine residues.**

| | | | ATOM 1 | | | | | | ATOM 2 | | | |
|---|---|---|---|---|---|---|---|---|---|---|---|---|
| | Atom no. | Atom name | Res name | Res no. | Chain | | Atom no. | Atom name | Res name | Res no. | Chain | Distance |
| **HYDROGEN BONDS** | | | | | | | | | | | | |
| 1 | 7 | OE1 | GLU | 27 | A | <—> | 6814 | N | GLY | 106 | B | 2.7 |
| 2 | 146 | O | GLU | 42 | A | <—> | 6625 | OG | SER | 81 | B | 3.27 |
| 3 | 294 | NZ | LYS | 57 | A | <—> | 7649 | O | CYS | 208 | B | 2.79 |
| 4 | 746 | NE2 | HIS | 103 | A | <—> | 7665 | O | TRP | 209 | B | 2.94 |
| 5 | 763 | OG | SER | 105 | A | <—> | 7716 | OD2 | ASP | 216 | B | 3 |
| 6 | 965 | NE2 | GLN | 129 | A | <—> | 7715 | OD1 | ASP | 216 | B | 2.75 |
| 7 | 3190 | NH1 | ARG | 355 | A | <—> | 11098 | OE1 | GLU | 592 | B | 2.88 |
| 8 | 3193 | NH2 | ARG | 355 | A | <—> | 11101 | O | GLU | 592 | B | 2.87 |
| 9 | 3193 | NH2 | ARG | 355 | A | <—> | 11098 | OE1 | GLU | 592 | B | 2.66 |
| 10 | 3383 | OE2 | GLU | 376 | A | <—> | 11076 | NZ | LYS | 590 | B | 2.6 |
| 11 | 4068 | NH2 | ARG | 447 | A | <—> | 9186 | O | ARG | 383 | B | 2.73 |
| 12 | 4305 | O | SER | 471 | A | <—> | 9264 | OG | SER | 392 | B | 3.18 |
| 13 | 4310 | OG | SER | 472 | A | <—> | 9179 | NH1 | ARG | 383 | B | 2.82 |
| 14 | 4521 | OE2 | GLU | 494 | A | <—> | 9216 | NH2 | ARG | 385 | B | 2.6 |
| 15 | 4542 | NH1 | ARG | 496 | A | <—> | 9235 | O | GLY | 387 | B | 3.17 |
| 16 | 4542 | NH1 | ARG | 496 | A | <—> | 9247 | O | GLY | 389 | B | 3.03 |
| 17 | 4542 | NH1 | ARG | 496 | A | <—> | 9254 | O | PRO | 390 | B | 2.75 |
| 18 | 4545 | NH2 | ARG | 496 | A | <—> | 9235 | O | GLY | 387 | B | 2.68 |
| 19 | 4555 | OD1 | ASN | 497 | A | <—> | 9179 | NH1 | ARG | 383 | B | 3.35 |
| 20 | 4555 | OD1 | ASN | 497 | A | <—> | 9182 | NH2 | ARG | 383 | B | 3 |
| 21 | 4555 | OD1 | ASN | 497 | A | <—> | 9264 | OG | SER | 392 | B | 3.15 |
| **SALT BRIDGES** | | | | | | | | | | | | |
| 1 | 3193 | NH2 | ARG | 355 | A | <—> | 11098 | OE1 | GLU | 592 | B | 2.66 |
| 2 | 3382 | OE1 | GLU | 376 | A | <—> | 11076 | NZ | LYS | 590 | B | 2.6 |
| 3 | 4520 | OE1 | GLU | 494 | A | <—> | 9216 | NH2 | ARG | 385 | B | 2.6 |

rigidity of the vaccine molecule is also a relevant factor that would be modulated by the linkers. An adjuvant was added in the N and C terminals of the vaccine as an immunomodulator and to enhance the activity of the vaccine. Also the vaccine was supported by 6His-tag molecule to facilitate the process of isolation and purification of the vaccine molecule [91].

Most importantly for the vaccine to be considered as better vaccine it should possess antigenic properties, which are important to elicit the immune response of the host. In addition to that it must not present homology with human proteins to escape the potentiality of causing autoimmune response [51,52,91–93]. In this study the designed vaccine demonstrated antigenicity in Vaxijen server (scored 0.6228 passing 0.4 the threshold of the server) and showed least homology to human proteins. These results guarantee that the vaccine could elicit the immune system without provoking autoimmune diseases and ease of the vaccine expression. Also, the immunologic analysis of vaccine structure demonstrated MHC-I, MHC-II epitopes accompanied by linear B-cell epitopes as previously described [93,94]. Most importantly the physio-chemical properties of the vaccine were analyzed and demonstrating that the vaccine protein was stable, contains aliphatic side chains, hydrophilic, no TMH and with thermal stability.

The study of the quality of proteins secondary and tertiary structures are of crucially significant for efficient presentation of antigenic peptides on MHC for triggering strong immune reactions [95]. In this study the determination of the quality

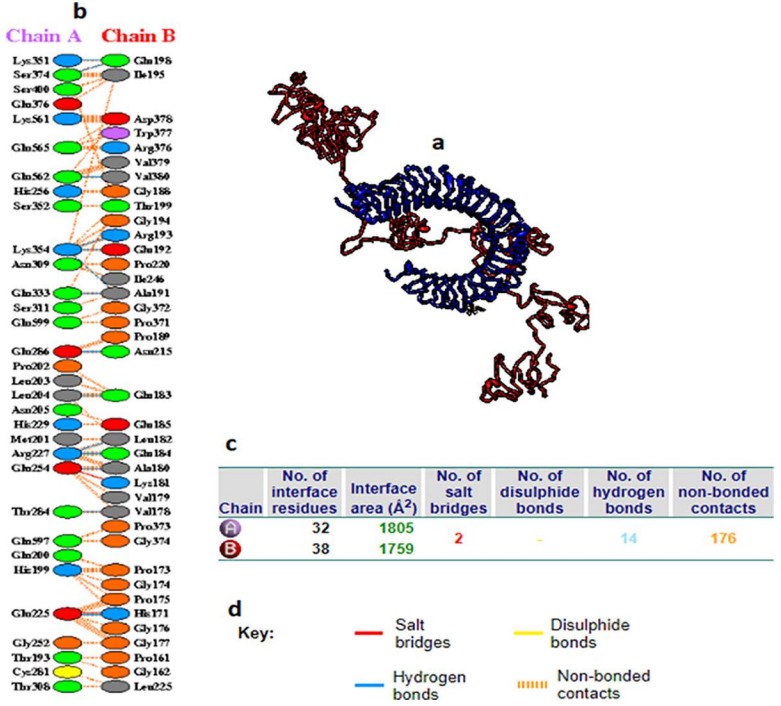

**Fig 11. (a) Docking of the vaccine construct (red color) with the TLR4 chain B (blue color).** (b) Residue interactions across the interface colored by residue type. The vaccine residues were presented as chain B and the TLR4 as chain A. (c) The interface statistics showing the number of bonded and unbonded residues. (d) The key showing the interacted bonds based on the color.

**Table 8. Atom−Atom Interactions list of the TLR4 chain B and the vaccine interface showing the hydrogen and salts bonding residues. Atom 1 represents TLR4 chain B and atom 2 represents the vaccine residues.**

| | | | ATOM 1 | | | | | | ATOM 2 | | | |
|---|---|---|---|---|---|---|---|---|---|---|---|---|
| | Atom no. | Atom name | Res name | Res no. | Chain | | Atom no. | Atom name | Res name | Res no. | Chain | Distance |
| **HYDROGEN BONDS** | | | | | | | | | | | | |
| 1 | 1733 | N | LEU | 204 | A | <—> | 7479 | O | GLN | 183 | B | 3.15 |
| 2 | 1935 | OE2 | GLU | 225 | A | <—> | 7491 | NE2 | HIS | 171 | B | 2.64 |
| 3 | 1959 | NH2 | ARG | 227 | A | <—> | 7398 | O | LEU | 182 | B | 2.7 |
| 4 | 1959 | NH2 | ARG | 227 | A | <—> | 7498 | OE1 | GLN | 184 | B | 2.64 |
| 5 | 2221 | OE1 | GLU | 254 | A | <—> | 7452 | N | ALA | 180 | B | 2.62 |
| 6 | 2513 | OG1 | THR | 284 | A | <—> | 7443 | O | VAL | 178 | B | 3.11 |
| 7 | 2534 | OE2 | GLU | 286 | A | <—> | 7746 | ND2 | ASN | 215 | B | 3.25 |
| 8 | 3006 | NE2 | GLN | 333 | A | <—> | 7548 | O | ALA | 191 | B | 2.83 |
| 9 | 3191 | O | LYS | 351 | A | <—> | 7615 | NE2 | GLN | 198 | B | 2.95 |
| 10 | 3216 | NZ | LYS | 354 | A | <—> | 7548 | O | ALA | 191 | B | 2.65 |
| 11 | 3216 | NZ | LYS | 354 | A | <—> | 7558 | O | GLU | 192 | B | 2.78 |
| 12 | 3216 | NZ | LYS | 354 | A | <—> | 7575 | O | ARG | 193 | B | 2.69 |
| 13 | 3404 | OG1 | SER | 374 | A | <—> | 7615 | NE2 | GLN | 198 | B | 2.99 |
| 14 | 5235 | NE2 | GLN | 562 | A | <—> | 9191 | O | VAL | 380 | B | 2.89 |
| **SALT BRIDGES** | | | | | | | | | | | | |
| 1 | 1934 | OE1 | GLU | 225 | A | <—> | 7398 | NE2 | HIS | 171 | B | 2.64 |
| 2 | 2222 | OE2 | GLU | 254 | A | <—> | 7465 | NZ | LYS | 181 | B | 3.35 |

**Table 9. The binding affinity between the TLR4-vaccine complexes. The binding affinity of the complexes was provided as ΔG and Kd. All the number of intermolecular contacts (ICs) and the percentage of the charged and a polar non-interacting surface (NIS%) of the complex were shown in the table.**

| Protein-protein complex | ΔG (kcal mol-1) | Kd (M) at 25 °C | ICs charged-charged | ICs charged-polar | ICs charged-a polar | ICs polar-polar | ICs polar-a polar | ICs a polar-a polar | NIS charged | NIS a polar |
|---|---|---|---|---|---|---|---|---|---|---|
| TLR4 chain A+vaccine | −19.2 | 4.90E-05 | 7 | 15 | 22 | 6 | 31 | 17 | 20.04 | 39.43 |
| TLR4 chain B+ vaccine | −21.8 | 3.20E-05 | 6 | 21 | 18 | 6 | 27 | 15 | 17.22 | 37.32 |

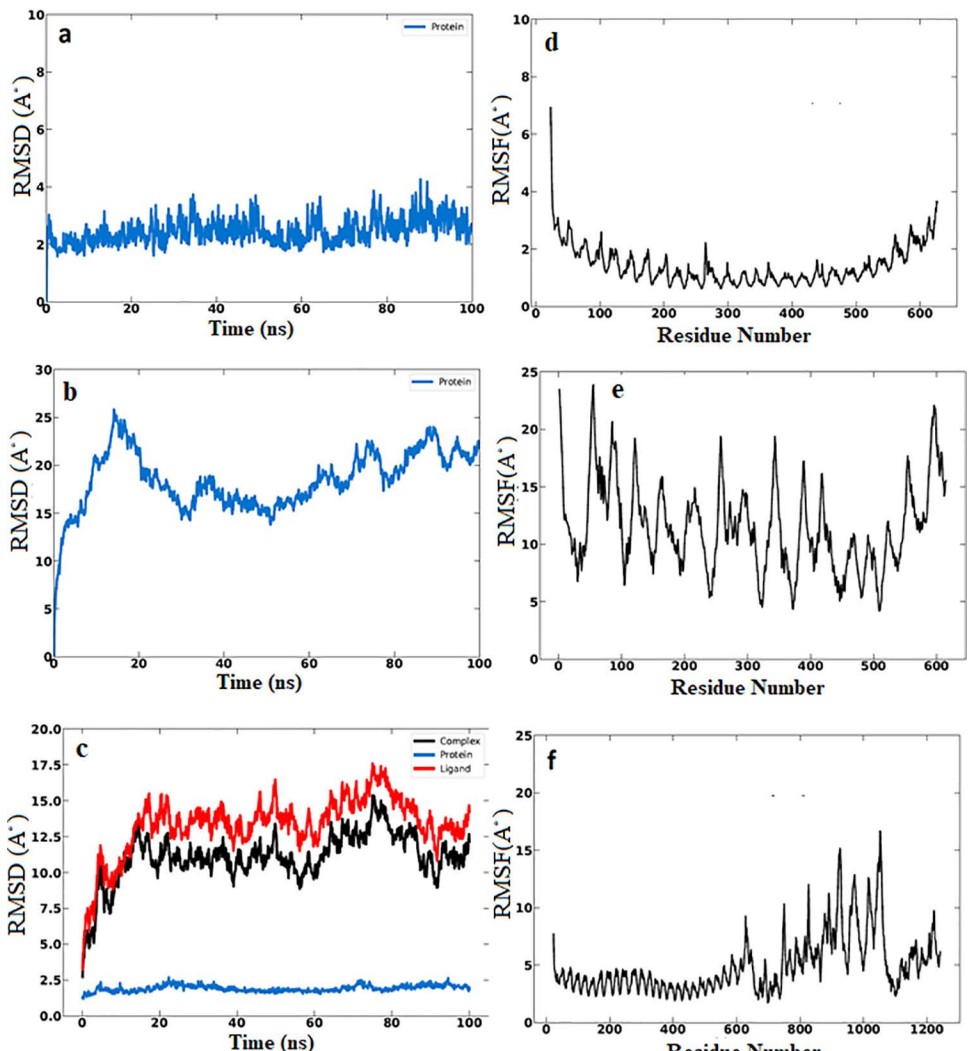

**Fig 12. The (a): RMSD of the TLR4 (b): RMSD of the vaccine construct.** (c): RMSD of the TLR4-vaccine complex. The RMSD of the demonstrate stability of the structures. For instance the RMSD plateaued around 2.5 to 5 Å after 20 ns, indicating the systems reached equilibrium. (d) The RMSF of TLR4. (e) RMSF of the vaccine molecule. (f) The RMSF of TLR4-vaccine complex.The RMSF identifies flexible regions in the molecules.

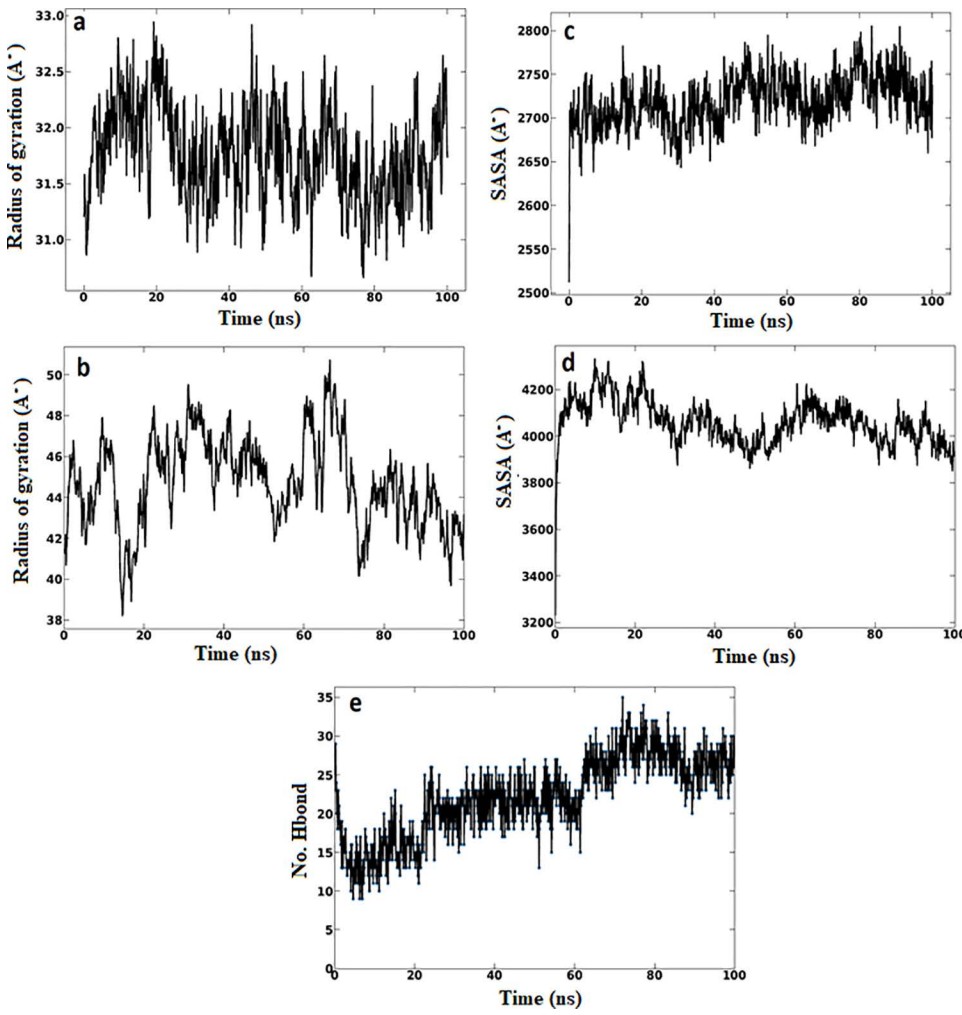

**Fig 13.** **(a) The radius of gyration of the TLR4.** (b) The radius of gyration of the vaccine molecule. (c) The SASA of the TLR4. (d) The SASA of the vaccine molecule. Radius of gyration showed the compactness of the TLR4 and the vaccine over time. The SASA plot demonstrated the vaccine molecule was stable and compact during the time of simulation. (e) The hydrogen bonding in the vaccine-TLR4 complex that strengthen the binding process of the interacting molecules.

and potential errors in the structural model of the vaccine were analyzed. Secondary structure showed the vaccine had helixes, beta strands and loops. These results provided the high solubility and polarity of the vaccine model structure. Tertiary structure was predicted, refined and assessed in ProSA web server. The model Z-score was −3.86, which falls within those commonly observed in similar size-native proteins [56]. Furthermore, the model was assessed by Ramachandran plot. The result showed that the vast amount of the residues were located in most-favored region and very few residues were in disallowed regions. This result remarkably demonstrated that the excellent quality and stability of the vaccine structure [96].

The analysis of the solubility of the vaccine is of great importance since the vaccine molecule works in liquid environment the host body. Silva, et al. previously reported that vaccines with least solubility were characterized with least production of viral proteins [97]. Thus, the solubility of the vaccine was assessed compared to the solubility of *E. coli* proteins using protein sol server [59,60]. The designed vaccine showed solubility scored of 0.580 greater than that of the of the

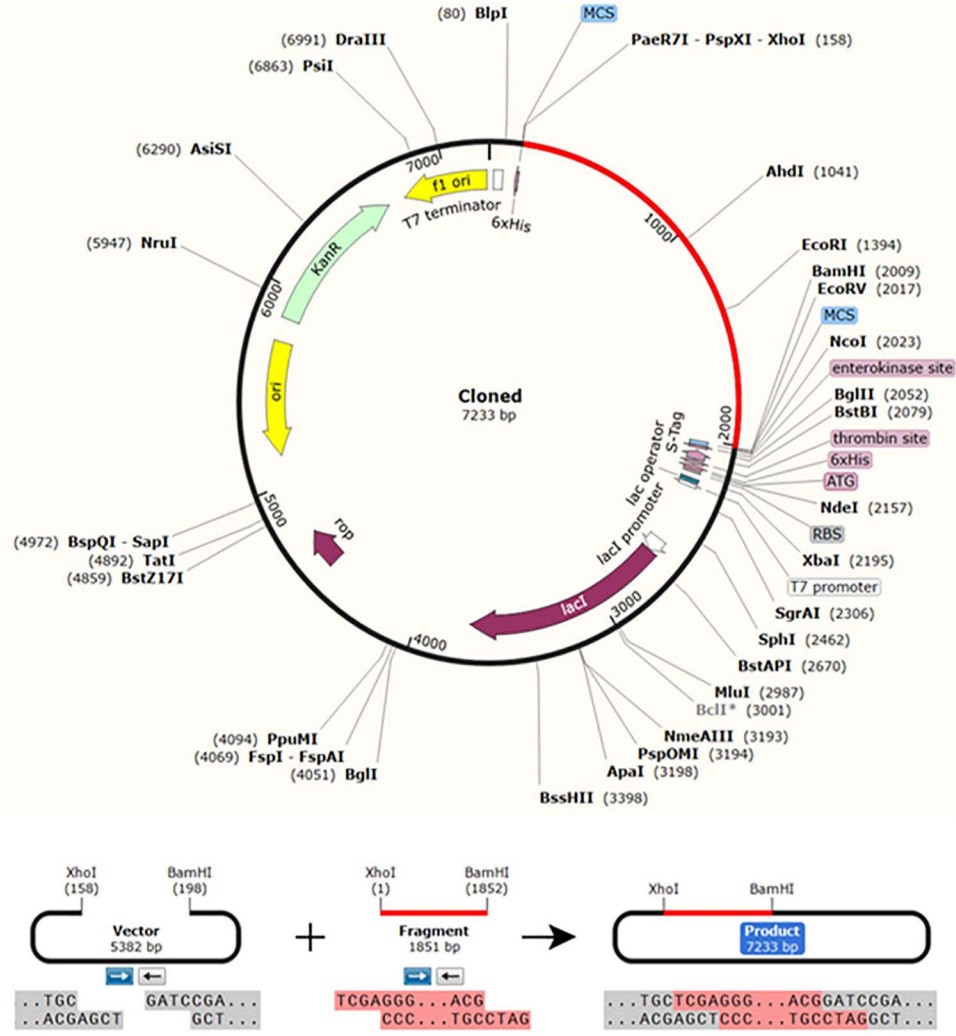

**Fig 14. *In silico* cloning of the final vaccine construct sequence in the pET28a ( +) expression vector.** The vector was shown in black color, while the red color provided the gene coding for the vaccine. The DNA sequence of the vaccine was typically cloned in the MCS of the vector between BamH1 and Xho1 cutting sites.

*E. coli* proteins (0.45). This result providing the good solubility of the vaccine protein in this study. Moreover the disulfide engineering is important for protein stability and proper folding. Besides that, structural disulfide bonding reduces the number of possible conformations for a given protein. This would result in remarkable reduction of the entropy and accompanied by increased thermostability [61,98]. Thus the stability of the vaccine construct was enhanced when six residues in the mobile regions were mutated to cysteine.

Results from the immunological stimulation were in line with actual immune responses. Immune responses were generally increased following repeated vaccination (antigen) exposure. Furthermore, memory cells in B and T lymphocytes showed remarkable growth. Most notably, after the first injection, IL-2 and IFN-γ levels increased and reached their maximum level with many exposures to the antigen, demonstrating high T-helper cell counts and effective immunoglobulin synthesis. The Simpson index revealed that the vaccine structure has a large number of B- and T-cell epitopes, suggesting a possible unique immune response [63,80].

Molecular docking is a computational tool predicting and calculating the binding affinity of the ligands to receptors. Here in this study, molecular docking was performed to validate the binding of the candidate epitopes or the vaccine to the immune receptors. Initially, each epitope was individually docked or interacted with the MHC-1 and MHC-II molecules to guarantee better interaction of the epitopes with host immune receptors. The binding of these epitopes to MHC molecules showed strong interactions based on the binding energies obtained and the hydrogen bond formation between the epitopes and the MHC molecules (Table 6, Figs 8 and 9). Secondly, the docking was performed between the vaccine molecule and the host TLR4. TLR4 is considered as an important receptor for stimuli against the infectious and/or non-infectious agents resulting in instigating the pro-inflammatory response and amplification of the inflammatory response [99]. Here the docking was performed using HADDOCK server since it relies on the use of biochemical and/or biophysical interaction data for the docking process. These data like the chemical shift perturbation data obtained from the experiments of NMR titration or mutagenesis data [100]. The majority of the other docking methods did not use the experimental data, instead, were based on a combination of energetics and shape complementary of the docked molecules. Therefore HADDOCK server was used to study the interaction of the vaccine with TLR4. The docked complex demonstrated lower binding energy scores (negative values) conferring the highest binding between the molecules. Furthermore, it is importantly to go deeper to understand the accurate prediction of the binding strength that driven protein-protein interaction. PRODIGY server is a simple but highly effective method that based on the intermolecular interactions and features derived from non-interface surface. Therefore the binding affinity of the complex (vaccine-TLR4) was assessed in terms of free energy change and constant of dissociation using the PRODIGY server. All the numbers of intermolecular contacts and the percentage of the charged and a polar non-interacting surface of the complex showed the energetic viability of the interacting molecules.

The stability of the TLR-vaccine complex was analyzed using MD simulation. The RMSD of the vaccine showed C-α atoms were slightly high. This might be attributed to the flexible, non-compact structure and high content of flexible loop regions. Nevertheless, the result signifying the potential role in stability of the vaccine structure. Also the RMSF result showed the minor fluctuation in the structure showing the rigidity of the structure. This result was previously recognized by Amin Rani et al.(2023) [101]. Moreover, it was previously mentioned that, high radius of gyration impacted the flexibility of the structure, providing less stable and compact structure [102]. In this study, the MD simulations demonstrated the TLR and the vaccine with lower Rg, indicating the compactness and stability of the structure. Moreover the hydrogen bonds interaction confer significant holding of the TLR-vaccine residues showing strong protein-protein interaction [103]. Therefore the more the hydrogen bonding interaction, the more stable protein-protein complex [104]. In this study the hydrogen bonds supporting the TLR4 chain A and chain B when each docked with the vaccine construct were 21 and 14 bonds, respectively. This result showed the potential stability and binding affinity of these complexes. In general the MD simulation demonstrated strong stability of TLR4-vaccine complexes.

Most importantly, the vaccine was molecularly cloned in a suitable cloning vector for immunoreactivity [77]. The best option for cloning was the *E. coli* expression system [105,106]. Codon usage optimization was performed to guarantee full production of the intended vaccine protein [106]. The high proportionate level of abundant codons was shown by the CAI of 1.0, and the high level of protein expression in bacteria was indicated by the GC-content of 57.929, which fell within the optimal range of 30% −70%.

Despite the promising results of the designed vaccine in this study, multiple challenges face the *in silico* vaccine development. For instance the variations gaps through populations and pathogen genomic data can hindered the prediction of the *in silico* models. Also the multi-omics data need expertise and significant computational analysis resources. Additionally, the complex immune responses prediction, like cellular immunity and long-term immune response, remains a challenge. Thus inadequate understanding of immune response variability may lead to adverse effects disproportionately affecting vulnerable groups.

## 5. Conclusion

In this study, the proposed epitopes for the development of an *in silico* vaccine against VZV were examined and analyzed. The epitopes were found to be immunogenic, non-allergenic and nontoxic. Moreover, the proposed vaccine successfully elicited B- and T-cells with excellent population coverage. Also the proposed vaccine was not implicated in autoimmunity for the host and showed good immune response by interacting with host immune system. Moreover, the vaccine molecule showed favorable interactions with human immune receptors such as MHC-I, MHC-II and TLR4. The proposed epitopes require applying *in vitro* or *in vivo* studies to evaluate their efficacy as efficient vaccine. This is because the proposed epitopes might carry unforeseen risks.

## Supporting information

**S1 Table. The 73 proteins of VZV, their Entries and lengths of each proteins from uniprot database.**
(DOCX)

**S2 Fig. The sequence alignment of the strain's sequences for (a): Tegument protein UL46 homolog.** (b): Capsid protein (c): Envelope glycoprotein C. (d): Envelope glycoprotein B (e): Capsid scaffolding protein.
(DOCX)

## Author contributions

**Conceptualization:** Yassir A. Almofti.

**Data curation:** Yassir A. Almofti, Amna A. Ibrahim, Nosiba Ibrahim, Nawal Elkhair, Sheryar Afzal, Ali Attiq, Yuan-Seng Wu.

**Formal analysis:** Yassir A. Almofti, Amna A. Ibrahim, Samir Alhojaily, Ali Attiq, Yuan-Seng Wu.

**Funding acquisition:** Ibrahim Albokhadaim.

**Investigation:** Amna A. Ibrahim, Abdelmajeed M. Elshafei, Nosiba Ibrahim, Saad Shousha, Samir Alhojaily.

**Methodology:** Yassir A. Almofti, Amna A. Ibrahim, Nuha A. Mahmoud, Sheryar Afzal.

**Project administration:** Yassir A. Almofti, Saad Shousha.

**Resources:** Ahmed O Alameen, Mohammed Ali Al-Hammadi.

**Software:** Amna A. Ibrahim, Nuha A. Mahmoud, Abdelmajeed M. Elshafei, Mahmoud Kandeel.

**Supervision:** Yassir A. Almofti.

**Validation:** Yassir A. Almofti, Ahmed O Alameen, Mahmoud Kandeel.

**Visualization:** Yassir A. Almofti, Abdelmajeed M. Elshafei, Nawal Elkhair.

**Writing – original draft:** Yassir A. Almofti, Nuha A. Mahmoud, Ibrahim Albokhadaim, Mohammed Ali Al-Hammadi, Ghada M.

**Writing – review & editing:** Yassir A. Almofti, Mahmoud G El Sebaei.

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
