## [Decision Letter · Decision Letter 0]

PONE-D-24-54052Exploring Varicella Zoster Virus Proteome for Construction and Validation of  A multi-Epitope Based Subunit Vaccine Using Multifaceted Immunoinformatics ApproachesPLOS ONE

Dear Dr. Almofti,

Thank you for submitting your manuscript to PLOS ONE. After careful consideration, we feel that it has merit but does not fully meet PLOS ONE’s publication criteria as it currently stands. Therefore, we invite you to submit a revised version of the manuscript that addresses the points raised during the review process by Reviewer #2. Please submit your revised manuscript by Apr 11 2025 11:59PM. If you will need more time than this to complete your revisions, please reply to this message or contact the journal office at plosone@plos.org . Please include the following items when submitting your revised manuscript:

We look forward to receiving your revised manuscript.

Kind regards,

Syed Nisar Hussain Bukhari

Academic Editor

PLOS ONE

Journal Requirements:

Reviewers' comments:

Reviewer's Responses to Questions

**Comments to the Author**

1. Is the manuscript technically sound, and do the data support the conclusions?

Reviewer #1: Yes

Reviewer #2: Yes

2. Has the statistical analysis been performed appropriately and rigorously? 

Reviewer #1: Yes

Reviewer #2: Yes

3. Have the authors made all data underlying the findings in their manuscript fully available?

Reviewer #1: Yes

Reviewer #2: Yes

4. Is the manuscript presented in an intelligible fashion and written in standard English?

Reviewer #1: Yes

Reviewer #2: Yes

5. Review Comments to the Author

Reviewer #1: Dear Authors

I sincerely appreciate the time and effort you have invested in addressing the comments. Your thorough responses and dedication to improving the quality of the work are commendable. Completing these revisions is a crucial step toward finalizing the manuscript, and your attention to detail has been invaluable.

Thank you once again for your commitment and hard work.

Best regards

Reviewer #2: The Manuscript covers an important topic in the world of bio-informatics and vaccine design using plenty of computational algorithms and in-silico analysis spreading over more than 40 pages making it text heavy.

The authors are advised to make their writing concise without compromising on the information contained in the manuscript as most of the information in Materials and Results when working with computational tools is repeated.

Because the paper is an experimental paper, the methodology section should contain a process flow diagram of the whole design process, which would make it intuitive for the reviewer to get a quick grasp of the whole work.

The manuscript has language and grammatical issues. It should be properly proof read and necessary changes should be incorporated to improve its readability before accepting it for publication.

The figures available in the manuscript do not seem to be publication level . Authors are advised to improve the quality of Figures.

A scientific study for validating the epitope characteristics of short listed VZV proteins should have been included so that the computational prediction has some relevance. Plain predictions, without any experimental support is prone to inaccuracies and is usually not counted as a result. Therefore, authors are advised to explore the validation literature for the epitope candidates which would greatly improve the authenticity of their results.

6. PLOS authors have the option to publish the peer review history of their article (what does this mean? ). If published, this will include your full peer review and any attached files.

**Do you want your identity to be public for this peer review?** For information about this choice, including consent withdrawal, please see our Privacy Policy .

Reviewer #1: No

Reviewer #2: No

---

## [Author Response · Author response to Decision Letter 1]

7 Apr 2025

Response to Editor and Reviewers comments

Dear Editor and Reviewers,

We sincerely appreciate the time and efforts that have been taken to provide insightful comments and suggestions. The feedback has been invaluable in improving the quality of our manuscript. Below, we provided a point-by-point response to each comment. Revisions made in the manuscript are highlighted in RED. The YELLOW colour is the removed parts, the repeated parts, in the result section as advised by the second reviewer.

Editor’s comments:

1. Comment:

Response:

Thank you very much for your valuable comment

We arranged our manuscript to meet the PLOS ONE's style requirements, including those for file naming.

2. Comment

Please note that PLOS ONE has specific guidelines on code sharing for submissions in which author-generated code underpins the findings in the manuscript. In these cases, we expect all author-generated code to be made available without restrictions upon publication of the work.

Response:

Thank you very much for your valuable comment

We provided the below link as available code to without restrictions.

(https://github.com/YassirAlmofti/vzvinsilicovac)

3. Comment

We note that the grant information you provided in the ‘Funding Information’ and ‘Financial Disclosure’ sections do not match.

Response:

Thank you very much for your valuable comment

Yes we agreed to your comment

We modified the grant information in the ‘Funding Information’ and ‘Financial Disclosure’ to be in the right format. Also we provided the correct grant number for the awards we received for our study in the ‘Funding Information’ section.

4. Comment

Please include captions for your Supporting Information files at the end of your manuscript, and update any in-text citations to match accordingly.

Response:

Thank you very much for your valuable comment

Yes we agreed to your comment

We have included captions for our Supporting Information files at the end of the manuscript and after the references list, and we updated citations to match accordingly in the manuscript.

5. Comment

Response:

Thank you very much for your valuable comment

Yes we agreed to your comment

We have reviewed the reference list to ensure that it is complete and correct. Moreover there are no retracted references cited in the manuscript.

Reviewer’s comments:

Reviewer #1:

Dear Authors:

I sincerely appreciate the time and effort you have invested in addressing the comments. Your thorough responses and dedication to improving the quality of the work are commendable. Completing these revisions is a crucial step toward finalizing the manuscript, and your attention to detail has been invaluable.

Thank you once again for your commitment and hard work.

Best regards

Response:

Thank you very much for your valuable response

Reviewer #2:

1. Comment:

The Manuscript covers an important topic in the world of bio-informatics and vaccine design using plenty of computational algorithms and in-silico analysis spreading over more than 40 pages making it text heavy.

The authors are advised to make their writing concise without compromising on the information contained in the manuscript as most of the information in Materials and Results when working with computational tools is repeated.

Response:

Thank you very much for your valuable comment

Yes we agreed to your comment as some parts in the materials and methods were repeated in the results section

We attempted to remove the repeated parts in the result section and rewrite them to concise without compromising the information contained. The removed repeated sections were highlighted with yellow to facilitate the follow up.

2. Comment

Because the paper is an experimental paper, the methodology section should contain a process flow diagram of the whole design process, which would make it intuitive for the reviewer to get a quick grasp of the whole work.

Response:

Thank you very much for your valuable comment

Yes we agreed to your comment

We added a flow diagram for the whole design process for the vaccine design. The flowchart was given Fig1, and thus numbering of the other Figures was revised.

3. Comment

The manuscript has language and grammatical issues. It should be properly proof read and necessary changes should be incorporated to improve its readability before accepting it for publication.

Response:

Thank you very much for your valuable comment

Yes we agreed to your comment

We had revised the manuscript grammatical errors and spelling mistakes for better readability.

4. Comment

The figures available in the manuscript do not seem to be publication level. Authors are advised to improve the quality of Figures.

Response:

Thank you very much for your valuable comment

Yes we agreed to your comment

We attempted to improve the quality of the figures to be in a better resolution

5. Comment

A scientific study for validating the epitope characteristics of short listed VZV proteins should have been included so that the computational prediction has some relevance. Plain predictions, without any experimental support is prone to inaccuracies and is usually not counted as a result. Therefore, authors are advised to explore the validation literature for the epitope candidates which would greatly improve the authenticity of their results.

Response:

Thank you very much for your valuable comment

We thank the reviewer for his valuable suggestion. While we acknowledged that experimental validation is essential for confirming the reliability of computational predictions, such validation is beyond the scope of the current study. However, we have now explicitly mentioned this as a limitation in the discussion section and emphasized the need for future experimental studies to validate the predicted epitopes. Additionally, we have explored relevant literature supporting the immunogenicity of similar epitopes. These references have now been incorporated into the manuscript in the discussion section (see ref: 81 and 101), strengthening the reliability of our findings.

---

## [Editor Report · Decision Letter 1]

Exploring Varicella Zoster Virus Proteome for Construction and Validation of  A multi-Epitope Based Subunit Vaccine Using Multifaceted Immunoinformatics Approaches

PONE-D-24-54052R1

Dear Dr. Almofti,

We’re pleased to inform you that your manuscript has been judged scientifically suitable for publication and will be formally accepted for publication once it meets all outstanding technical requirements.

Kind regards,

Syed Nisar Hussain Bukhari

Academic Editor

PLOS ONE
---

## [Editor Report · Acceptance letter]

PONE-D-24-54052R1

PLOS ONE

Dear Dr. Almofti,

I'm pleased to inform you that your manuscript has been deemed suitable for publication in PLOS ONE. Congratulations! Your manuscript is now being handed over to our production team.

Kind regards,

on behalf of

Dr. Syed Nisar Hussain Bukhari

Academic Editor

PLOS ONE